# *Lactobacillus paracasei* Expressing Porcine Trefoil Factor 3 and Epidermal Growth Factor: A Novel Approach for Superior Mucosal Repair

**DOI:** 10.3390/vetsci12040365

**Published:** 2025-04-14

**Authors:** Fangjie Yin, Ying Chen, Huijun Zhang, Hongzhe Zhao, Xuenan Li, Zi Wang, Weijing Meng, Jie Zhao, Lijie Tang, Yijing Li, Jiaxuan Li, Xiaona Wang

**Affiliations:** 1College of Veterinary Medicine, Northeast Agricultural University, Harbin 150030, China; 18754378593@163.com (F.Y.); 15662788212@163.com (Y.C.); zhanghuijun2021@163.com (H.Z.); hongzhezhao@163.com (H.Z.); lixuenan@neau.edu.cn (X.L.); tanglijie@163.com (L.T.); yijingli@163.com (Y.L.); 2Heilongjiang Key Laboratory for Animal Disease Control and Pharmaceutical Development, Harbin 150030, China; 3Tongliao Institute of Animal Husbandry and Veterinary Science, Tongliao 028000, China; wangzigaoxue@163.com; 4Tongliao Agricultural and Animal Husbandry Development Center, Tongliao 028000, China; tongliaoxiaomeng@163.com; 5Nanjing Dr. Vet Health Management Co., Ltd., Nanjing 210000, China; zhaojie_dr_vet@163.com

**Keywords:** porcine trefoil factor 3, porcine epidermal growth factor, *Lactobacillus paracasei*, inflammatory bowel disease, oral vaccine

## Abstract

Inflammatory bowel disease significantly impairs patients’ physical health and quality of life, presenting a major challenge in modern healthcare. This chronic condition, characterized by persistent gastrointestinal inflammation, not only affects physiological functions but also imposes considerable psychological and socioeconomic burdens on affected individuals and healthcare systems. The primary objective of this study was to investigate the therapeutic potential of orally administered porcine trefoil factor 3 and epidermal growth factor delivered by *Lactobacillus paracasei 27-2* for intestinal mucosal repair. Our experimental results demonstrated that this novel delivery system significantly enhanced the proliferation and migration capabilities of Immortalized Porcine Enterocyte Cell line J2, suggesting its promising application in promoting intestinal mucosal regeneration and repair. Moreover, the oral administration of the recombinant *L. paracasei 27-2* strains could effectively alleviate the symptoms of colitis induced by dextran sulfate sodium in mice and significantly improve the integrity of the intestinal mucosa. In conclusion, using *Lactobacillus paracasei 27-2* as an oral delivery vector to express trefoil factor and epidermal growth factor shows great potential as a novel strategy for treating inflammatory bowel disease.

## 1. Introduction

Inflammatory bowel disease (IBD) encompasses a spectrum of chronic, relapsing inflammatory conditions of the gastrointestinal tract that have emerged as a major global health concern, imposing significant socioeconomic burdens and presenting complex challenges to healthcare systems worldwide. The development of reliable animal models for inflammatory bowel disease (IBD) is crucial for advancing our understanding of its complex etiology and pathophysiological mechanisms. These models serve as indispensable tools for investigating disease progression and evaluating novel therapeutic interventions. Among various approaches, chemically induced IBD models, particularly those employing dextran sulfate sodium (DSS), trinitrobenzene sulfonic acid (TNBS), oxazolone, and acetic acid, have become the gold standard in preclinical research due to their well-established reliability, reproducibility, cost-effectiveness, and standardized protocols. Furthermore, the restoration of intestinal mucosal integrity has emerged as a critical endpoint in assessing disease remission and therapeutic efficacy, providing valuable insights into mucosal healing processes. Advances in IBD research and the introduction of new pharmacological agents have transformed the therapeutic focus from merely relieving symptoms to promoting mucosal repair and healing. Epithelial repair is regulated by numerous signaling molecules, such as epidermal growth factor (EGF) and intestinal trefoil factor (TFF) [1]. A small-scale short-term clinical trial has demonstrated the effectiveness of EGF in treating patients with IBD, with more than 80% of the patients achieving clinical remission [2]. As early as 2004, researchers used *lactic acid bacteria* (LAB) as a delivery vector to secrete intestinal trefoil factor. This approach effectively treated acute and chronic colitis as well as epithelial injury and proposed a novel treatment approach for IBD [3].

Trefoil peptides, a family of evolutionarily conserved bioactive molecules, have garnered significant scientific attention due to their essential functions in gastrointestinal mucosal protection and repair. In mammalian systems, these peptides are categorized into three structurally and functionally distinct subtypes: trefoil factor 1 (TFF1), which has been implicated in breast cancer progression; trefoil factor 2 (TFF2), characterized by its potent antispasmodic and mucosal protective properties; and trefoil factor 3 (TFF3), predominantly expressed in intestinal goblet cells and exhibiting extensive distribution throughout the intestinal epithelium. TFF3 exhibits structural conservation and contains anti-acidic, anti-proteolytic, and anti-thermal breakdown characteristics [4]. Intestinal goblet cells secrete TFF3 in conjunction with mucins, forming covalently linked multimers that constitute the fundamental structural components of the intestinal mucosal barrier. This barrier serves as a critical defense mechanism, maintaining intestinal homeostasis and providing protection against pathogenic invasion and luminal insults. Furthermore, TFF3 plays a pivotal role in mucosal repair processes by facilitating the migration and restitution of intestinal epithelial cells, thereby promoting the regeneration of damaged mucosal surfaces.

Extensive research has demonstrated that epidermal growth factor (EGF) serves as a critical regulator of fundamental cellular processes, including proliferation, differentiation, and programmed cell death (apoptosis). Particularly in the gastrointestinal tract, EGF has been shown to significantly enhance mucosal repair mechanisms through its potent mitogenic effects on epithelial cells, promoting both their growth and differentiation, which are essential for maintaining intestinal barrier integrity and function [5]. Studies have shown that EGF facilitates the migration and proliferation of epithelial cells during wound healing, therefore assisting in the safeguarding of the wound site from disturbance and infection [6]. The synergistic effects of TFF3 and EGF in promoting vascular proliferation have been demonstrated across various injury models. Evidence suggests that TFF3 works in conjunction with EGF to facilitate injury repair in in vitro cell scratch assays [7]. Additionally, studies conducted by Chinery et al. [8] have indicated that intestinal trefoil factor and epidermal growth factor exhibit similar synergistic effects in mitigating damage. In addition to its well-established protective functions, intestinal trefoil factor plays a pivotal role in mucosal healing and tissue repair processes. The impairment of intestinal barrier function has been strongly implicated in the pathogenesis of various gastrointestinal disorders, particularly through mechanisms involving increased intestinal permeability. This pathophysiological alteration is closely associated with the development of multiple gastrointestinal conditions, including food allergies, infectious enterocolitis, and IBD [9].

In recent decades, significant advancements have been made in scientific research on LAB and their associated health benefits. Consequently, the use of genetically modified LAB as mucosal delivery systems offers numerous advantages, including enhanced safety profiles that enable the production of foreign proteins without the need for purification. This approach allows for direct oral administration, thereby improving immunotherapeutic efficacy within the intestinal environment. *Lactobacillus* is recognized as a significant probiotic in the gastrointestinal tract, promoting gastrointestinal health and strengthening the immune system. The capacity to endure acidic conditions and bile salts, together with their adhesive characteristics, promotes the colonization, multiplication, and predominance of *Lactobacillus* in the gut [10]. This dominance plays a critical role in suppressing the proliferation of pathogenic microorganisms and sustaining the equilibrium of the gut microbiota [11]. In this study, *Lactobacillus paracasei 27-2*, a strain isolated from the intestinal tract of piglets, was employed as the host organism. To enhance its probiotic efficacy, the strain was genetically modified to express TFF3 and EGF, two bioactive molecules known to promote intestinal health and epithelial repair. Owing to its remarkable capacity for intestinal colonization and stress resistance, *L. paracasei 27-2* serves as an excellent vector for exogenous protein expression [12].

This work generated recombinant *L. paracasei 27-2* strains expressing pTFF3 and pEGF and assessed their efficacy in preventing and treating intestinal mucosal damage and IBD.

## 2. Materials and Methods

### 2.1. The Bacteria and Plasmid

The wild *Lactobacillus paracasei 27-2* strain, obtained from the intestines of big landrace pigs, was grown in de Man, Rogosa, and Sharpe (MRS) broth at 37 °C without shaking. The expression plasmid pPG-T7g10-PPT was synthesized in our laboratory [13]. Six-week-old SPF BALB/c mice were procured from Liaoning Changsheng Biotechnology Co., Ltd. (Liaoning, China).

### 2.2. Construction of Recombinant Lactobacillus

Initially, the target segments pEGF and pTFF3 and the fusion fragment pTE (comprising pEGF and pTFF3) were ligated into the pMD19T plasmid, subsequently cloning the amplicon into the expression vector pPG-T7g10-PPT to produce pPG-pTFF3, pPG-pEGF, and pPG-pTE. Following electroporation into porcine *L. paracasei 27-2*, three recombinant strains were generated and named pPG-pTFF3/*27-2*, pPG- pEGF/*27-2*, and pPG-pTE/*27-2* (flag tags were added to pTFF3, pEGF, and pTE, enabling their detection using anti-FLAG antibodies).

### 2.3. Protein Expression

The recombinant *L. paracasei 27-2* strains were cultivated in liquid medium enriched with chloramphenicol at 37 °C for 16 h and then subjected to centrifugation at 12,000× *g* for 5 min. Thereafter, the supernatant of the recombinant strains culture was harvested and subjected to trichloroacetic acid (TCA) treatment to promote protein precipitation. The protein particles acquired from low-speed centrifugation were further treated with acetone. Subsequently, Western blot analysis was conducted on the produced samples. Subsequent to electrophoresis, the proteins were transferred to a PVDF membrane (Millipore, Milford, MA, USA). The mouse anti-FLAG monoclonal antibody (Abmart, Shanghai, China), diluted 1:1000 in monoclonal antibody diluent, acted as the primary antibody, whereas goat anti-mouse IgG conjugated with horseradish peroxidase (HRP) (Sigma, Ronkonkoma, NY, USA), diluted 1:5000 in skim milk, served as the secondary antibody. Thereafter, the PVDF membrane (Millipore, Milford, MA, USA) was subjected to chemiluminescence detection using a chemiluminescence reagent (Thermo Scientific, Durham, NC, USA).

In addition, indirect immunofluorescence experiments were conducted to enhance the expression of target proteins in pPG/*27-2*, pPG-pTFF3/*27-2*, pPG-pEGF/*27-2*, and pPG-pTE/*27-2* strains. The technique is delineated as follows: An overnight culture of recombinant strains was cultivated for 16 h in MRS broth with chloramphenicol (10 µg mL^−1^), followed by centrifugation at 12,000× *g* for 5 min. The mouse anti-flag monoclonal antibody (Abmart, Shanghai, China) was used at a dilution of 1:2000, in conjunction with fluorescein isothiocyanate (FITC)-conjugated goat anti-mouse IgG (Invitrogen, Carlsbad, CA, USA) diluted at 1:4500, and incubated at ambient temperature for 3 h. Following this, the samples were washed three times with sterile phosphate-buffered saline (PBS) and then incubated with 4′,6′-diamino-2-phenylindole (DAPI) (Invitrogen, Carlsbad, CA, USA) for 5 min. Afterward, the samples were washed three times with sterile PBS, dehydrated with water, and analyzed using a fluorescence microscope (Zeiss, Oberkochen, Germany).

The recombinant *L. paracasei 27-2* strains that had been activated by streaking were inoculated into MRS broth with chloramphenicol (10 µg mL^−1^). When the optical density at 600 nm (OD600) of each group of recombinant *L. paracasei 27-2* strains reached 1, they were introduced into MRS medium at a 1:100 dilution to evaluate the expression levels of the target protein during further culture. Samples were collected at 6, 10, 14, 18, 22, and 24 h intervals to measure the quantity of target proteins in both the supernatant and bacterial lysate at these time points using commercial pTFF3 and pEGF enzyme-linked immunosorbent assay (ELISA) kits (Wuhan Enzyme Immunoassay Biotechnology Co., Ltd., Wuhan, China). The concentrations of pTFF3 and pEGF in the fusion protein were quantitatively determined using specific ELISA kits for each target protein (pTFF3 ELISA kit and pEGF ELISA kit, respectively). The experimental values were calculated by summing the mean values obtained from the pTFF3 and pEGF measurements. All experiments were performed in triplicate to ensure statistical reliability. The operational stages followed the methods specified in the ELISA kit, including the creation of standard curves to enable precise estimations of protein expression levels.

### 2.4. Growth Characteristics and Expression Stability

To assess the expression stability of the gene that was inserted, the recombinant *L. paracasei 27-2* strains were streaked on MRS plates and subjected to serial passage for 20 generations. The bacterial solution was inoculated into the liquid medium at a 1% dilution and incubated at 37 °C for 24 h. Samples of the bacterial suspension were collected every two hours to evaluate the viable bacterial count using the plate counting method, and thereafter the bacterial growth curve was plotted.

The recombinant *L. paracasei 27-2* strains were subjected to serial passage and culture over 20 generations. The plasmids extracted from the 5th, 10th, 15th, and 20th generations of each *L. paracasei 27-2* strain were then used as templates for PCR identification using the primer pair pPG-F/R. Simultaneously, recombinant *L. paracasei 27-2* strains from the 5th, 10th, 15th, and 20th generations were prepared to provide protein samples for assessing the expression stability of the proteins.

### 2.5. Analysis of the Activity of pTFF3 and pEGF In Vitro

Following the activation of the recombinant *L. paracasei 27-2* strains, the culture medium’s supernatant was gathered, and the proteins were extracted using the ammonium sulfate precipitation technique [14]. The ammonium sulfate precipitation was performed at ambient temperature (25 °C) through sequential addition of solid ammonium sulfate. The saturation levels were progressively increased in a stepwise manner, first to 20%, then to 40%, and finally to 50% saturation. By implementing a gradual salt concentration gradient, graded precipitation minimizes protein co-precipitation and enables more precise separation of target proteins based on their differential solubility characteristics [15]. The precipitated protein solution was centrifuged at 10,000× *g* for 20 min. The resulting precipitate was collected and dissolved in an appropriate volume of PBS. The dissolved proteins were then transferred into a dialysis bag and subjected to a 24 h dialysis procedure. Following dialysis, the sample was centrifuged at 10,000× *g* for 5 min at 4 °C to remove insoluble precipitates, and the resulting supernatant was carefully collected for subsequent analysis. Subsequently, the protein concentration in the supernatant was measured using pTFF3 and pEGF ELISA kits. (Wuhan Enzyme Immunoassay Biotechnology Co., Ltd., Wuhan, China). Subsequently, the supernatant was subjected to filtration using a sterile 0.22 µm filter and preserved at −80 °C.

The proliferative impacts of pTFF3, pEGF, and pTE on IPEC-J2 were evaluated using the CCK-8 test [7]. The cell density was adjusted to 2.0 × 10⁵/mL during cell passage. Then, 100 μL of cell suspension was added to each well of a 96-well plate. Subsequently, the plate with the added cell suspension was incubated in a moist environment containing 5% CO_2_ for 12 h. Following the removal of the culture medium, the wells were washed with PBS. Then, 100 µL of protein samples at different concentrations (0, 12.5 ng/mL (5.68 × 10^−3^ µg/mg), 25 ng/mL (1.14 × 10^−2^ µg/mg), 50 ng/mL (2.27 × 10^−2^ µg/mg), 100 ng/mL (4.54 × 10^−2^ µg/mg), 200 ng/mL (9.08 × 10^−2^ µg/mg), and 400 ng/mL (1.82 × 10^−1^ µg/mg)) was added to each well, with six replicates for each sample. Subsequently, 10 µL of CCK-8 solution (Sangon Biotech, Shanghai, China) was added to each well, and the cultures were incubated in the dark. Absorbance at 450 nm was recorded from each well every 30 min using an ELISA reader until the readings reached the optimum range for CCK-8 (the whole process takes approximately 4 h).

A cell scratch test was performed to evaluate the influence of pTFF3, pEGF, and pTE on the migratory ability of IPEC-J2 cells [16]. The IPEC-J2 cell suspension was seeded onto the culture plate until 80% confluence was achieved. A sterile 10 µL pipette tip was then used to create linear scratches in the cell monolayer. Using the optimal concentrations of each protein for cell proliferation (pPG: 100 ng/mL (total protein concentration); pEGF: 50 ng/mL (2.27 × 10^−2^ µg/mg); pTFF3: 100 ng/mL (4.54 × 10^−2^ µg/mg); pTE: 25 ng/mL (1.14 × 10^−2^ µg/mg)), the above-mentioned proteins were added, respectively, after the cell scratch assay. The healing area of the scratch was observed under an inverted fluorescence microscope at 0 h, 12 h, and 24 h.

### 2.6. Animal Model

Thirty-two six-week-old BALB/c mice were divided randomly into four groups, with each group containing eight mice and housed separately in cages. All experimental and animal management techniques received clearance from the Animal Experiment Institution Committee of Northeast Agricultural University in Harbin, China (approval number: 2016 NEFU-315, dated 13 April 2017). The individuals were acclimated to a controlled environment with a temperature of 22 ± 2 °C, consisting of 12 h of light and 12 h of dark. After a one-week acclimatization phase, food and water were provided ad libitum to the mice. The colitis model was established following the technique outlined by Wirtz [17], in which the mice were allowed unrestricted access to a 2% DSS solution for seven days. Oral recombinant *L. paracasei 27-2* strains: From day 1 to day 14 of the model establishment period, each mouse in this group was orally administered 200 μL (2 × 10^9^ CFU) of recombinant *L. paracasei 27-2* strains daily. Meanwhile, from the 8th day to the 14th day, each mouse was allowed to freely drink a 2% DSS solution every day. PBS group: Mice in this group were orally fed 200 μL of PBS daily from day 1 to day 14. Following the conclusion of this time, the mice were killed. Figure 1 depicts the experimental groups together with their respective feeding dosages, which are further elaborated in Table 1.

During the modeling phase, the body weight of the mice was recorded daily, and the characteristics of their feces were observed to check for the presence of diarrhea. Fecal samples were collected, and a fecal occult blood test kit (Baso, Guangdong, China) was used to detect occult blood in the feces. The disease activity index (DAI) score was computed using the Murthy scoring system [18], with detailed assessment criteria included in Table 2. On the eighth day after DSS therapy, blood samples were obtained from the ocular area of the mice in each experimental group. After euthanasia, the entire colon, from the cecum to the anus, was meticulously dissected from each mouse, and its length was accurately measured.

### 2.7. H&E Staining

After the completion of the modeling, a section of the colon (approximately 1 cm in length) near the rectum was excised from the mice of each group for hematoxylin and eosin (H&E) staining. A piece of colon tissue was preserved in 4% paraformaldehyde for 48 h. Subsequently, the tissue underwent dehydration by an ethanol gradient procedure with concentrations of 75%, 85%, 95%, and 100%. Following dehydration, the tissue was subjected to two 10 min washes with xylene. The tissue was then immersed in paraffin wax three times at a temperature of 57 °C for 60 min for each immersion. The colon specimens underwent a comprehensive processing protocol to prepare them for H&E staining, which included slide retrieval, baking, hematoxylin staining, washing, eosin counterstaining, dehydration, transparency treatment, and sealing [19]. The integrity of the colonic mucosal structure was observed, the crypt and glandular structures were compared under a light microscope, and the pathological damage was evaluated.

### 2.8. Myeloperoxidase (MPO) Activity Assay

Myeloperoxidase (MPO) activity in colonic tissue was evaluated. Tissue was weighed, and a 5% homogenate (1:19 w/v in PBS) was prepared. MPO activity in colon samples was measured using an ELISA kit (Nanjing Jiancheng Biotechnology Co., Ltd., Nanjing, China) according to the manufacturer’s instructions. The results were expressed as MPO activity units per gram of tissue, based on absorbance at 450 nm.

### 2.9. Gene Expression Levels in Colon Tissue

Gene expression levels in colon tissue were analyzed by real-time quantitative RT-PCR (qRT-PCR) and Western blot. Briefly, approximately 0.1 g of mouse colon tissue from each experimental group was homogenized in 1 mL of sterile PBS at a speed of 12,000× *g* for 2 min. Subsequently, RNA was extracted from the supernatant and subjected to reverse transcription. The target genes ZO-1, Claudin-2, and Occludin, along with the internal reference gene β-actin, were then quantified, and the data were analyzed using the 2^−△△Ct^ method.

The following steps were carried out: An equal quantity of mouse colon tissue was taken from each group. Then, 10 µL of PMSF was added, and the volume was adjusted to 1 mL with IP lysis buffer. Next, the samples were incubated on a shaker at 4 °C for 4 h, centrifuged at 5000× *g* at 4 °C for 5 min, and the supernatant was collected. The BCA protein detection kit (Beyotime, Shanghai, China) was utilized for protein measurement. It was ensured that protein concentrations were uniform across all groups and that the techniques specified in the previous section were adhered to. For Western blot analysis of the target protein, Claudin-2 (Servicebio, Wuhan, China), ZO-1, and Occludin (Proteintech, Wuhan, China) were chosen as the primary antibodies, which were used at a dilution of 1:500. Meanwhile, goat anti-mouse IgG conjugated with horseradish peroxidase (HRP) (Sigma, Ronkonkoma, NY, USA), diluted 1:5000 in skim milk, was used as the secondary antibody.

### 2.10. Cytokine Detection

On the 14th day after oral administration of recombinant *L. paracasei 27-2* strains, mouse serum samples were collected, and the cytokine levels of IL-1β, IL-6, IL-10, and TNF-α were evaluated using ELISA kits (Wuhan Enzyme Immunoassay Biotechnology Co., Ltd., Wuhan, China). Subsequently, a standard curve was constructed, and the associated cytokine concentrations were calculated.

### 2.11. Statistical Analysis

All the experiments in this study were repeated three times. The data were statistically analyzed using the two-way ANOVA method with the Graphpad Prism 8.0 software. As shown in the significant difference analysis of *p*-values, * *p* < 0.05, 0.01 < ** *p* < 0.05, and *** *p* < 0.01, while ns represents *p* > 0.05, and # represents the same significance range as *.

Values indicate the mean ± standard deviation (*n* = 3); bolded font indicates maximum values; different letters indicate significant differences between groups (*p* < 0.05); the same letters indicate non-significant differences (*p* > 0.05).

## 3. Results

### 3.1. Construction and Characterization of Recombinant Lactobacillus Expressing pTFF3 and pEGF

Subsequent to the confirmation of the PCR and sequencing findings, the strains pPG-pEGF/*27-2*, pPG-pTFF3/*27-2*, and pPG-pTE/*27-2* were analyzed (Figure 2a and Appendix A). The recombinant *L. paracasei 27-2* strains pPG-pTFF3/*27-2* (Figure 2b and Appendix A), pPG-pEGF/*27-2* (Figure 2c and Appendix A), and pPG-pTE/*27-2* (Figure 2d and Appendix A) displayed target proteins with molecular weights of about 13 kDa, 12 kDa, and 23 kDa, respectively, consistent with the anticipated sizes. The recombinant *L. paracasei 27-2* strains had unique short rod-shaped morphologies and demonstrated pronounced green fluorescence, unlike the control group, which revealed no luminescence. This discovery further substantiates the expression of the target proteins in the recombinant *L. paracasei 27-2* strains pPG-pTFF3/*27-2*, pPG-pEGF/*27-2*, and pPG-pTE/*27-2* (Figure 3).

Using commercially available pTFF3 and pEGF ELISA kits, standard curves were generated to measure the protein expression levels of each recombinant *L. paracasei 27-2* strains at multiple time points (Table 3 and Table 4). The recombinant *L. paracasei 27-2* strains exhibited maximum expression levels of the target proteins in both the bacterial supernatant and cells at 18 h, with expression levels in the bacterial cells significantly above those in the supernatant (*p* < 0.05).

### 3.2. Growth Characteristics and Stability Analysis of Recombinant L. paracasei 27-2 Strains

A thorough examination was performed to analyze the growth kinetics of recombinant strains. The results demonstrated that all three recombinant strains commenced the logarithmic growth phase at 4 h post-inoculation with the wild strain, eventually reaching a stable phase at approximately 16 h. Compared to the *L. paracasei 27-2*, their growth curve exhibits an S-shape, with consistent growth trends and no statistically significant changes in key growth characteristics such as growth rate or lag phase duration (Figure 4d). The recombinant *L. paracasei 27-2* strains were continuously propagated for 20 generations, plasmids every five generations, and then they were sequenced (Figure 4a–c, Appendix A). The integrity of the recombinant plasmids and their steady inheritance were validated using PCR and sequencing.

Additionally, stability analysis was performed on protein samples derived from the five generations of recombinant *L. paracasei 27-2* strains. Western blot examination demonstrated distinct bands at about 12 kDa, 13 kDa, and 23 kDa (Figure 4e–g, Appendix A), indicating stable production of the recombinant protein.

### 3.3. Examination of Cellular Proliferation in Relation to Specific Target Proteins

The results were analyzed by the CCK-8 method (Figure 5). pTFF3 and pEGF at 25 ng/mL (1.14 × 10^−2^ µg/mg), 50 ng/mL (2.27 × 10^−2^ µg/mg), 100 ng/mL (4.54 × 10^−2^ µg/mg), and 200 ng/mL (9.08 × 10^−2^ µg/mg) could significantly promote the cell proliferation of IPEC-J2, with the best proliferative effect of pTFF3 and pEGF at the concentrations of 100 ng/mL (4.54 × 10^−2^ µg/mg) and 50 ng/mL (2.27 × 10^−2^ µg/mg), respectively. It was found that 12.5 ng/mL (5.68 × 10^−3^ µg/mg), 25 ng/mL (1.14 × 10^−2^ µg/mg), 50 ng/mL (2.27 × 10^−2^ µg/mg), 100 ng/mL (4.54 × 10^−2^ µg/mg), and 200 ng/mL (9.08 × 10^−2^ µg/mg) of pTE could greatly enhance the growth of IPEC-J2 cells, in which the best proliferation effect was observed at 25 ng/mL (1.14 × 10^−2^ µg/mg). When the stimulus concentration is low, the number of receptors on the cell surface that bind to the stimulus is limited, and the activated intracellular signaling pathways are relatively weak. With the increase in stimulation concentration, more receptors are activated, and the signaling pathway gradually strengthens, thereby promoting cell proliferation, and the proliferation rate shows an upward trend. Based on this characteristic, the proliferation effect of these factors on cells can be weakened by inhibiting their receptors [20,21,22]. When the stimulus concentration reaches a certain level, signaling molecules are in a state of maximum activation. At this point, continuing to increase the stimulation concentration may lead to interruption or overactivation of signaling pathways, triggering intracellular stress responses and inhibiting cell proliferation [23,24,25].

### 3.4. Analysis of the Effects of pTFF3, pEGF, and pTE on Cell Migration

The influence of pTFF3, pEGF, and pTE proteins on the migration of IPEC-J2 cells was assessed through a cell scratch assay. Using the optimal concentrations of each protein for cell proliferation (pPG: 100 ng/mL (total protein concentration); pEGF: 50 ng/mL (2.27 × 10^−2^ µg/mg); pTFF3: 100 ng/mL (4.54 × 10^−2^ µg/mg); pTE: 25 ng/mL (1.14 × 10^−2^ µg/mg); pEGF + pTFF3: pEGF—50 ng/mL (2.27 × 10^−2^ µg/mg), pTFF3—100 ng/mL (4.54 × 10^−2^ µg/mg)), the above-mentioned proteins were added, respectively, after the cell scratch assay. Observations revealed that the cell scratches in all experimental groups began to close after 12 h of protein treatment, with nearly complete closure observed in the pTFF3 and pTE groups after 24 h. The extent of cell migration at 24 h was quantified using ImageJ 1.53t software (https://imagej.net/ij, accessed on 4 October 2024). Compared to the pPG group, significant enhancements in cell migration were noted in the pTFF3, pEGF, pTFF3 + pEGF, and pTE groups. Remarkably, the effect detected in the pTE group was significantly more prominent compared to that observed in either the pTFF3 or the pEGF group. However, there was no significant difference between the pEGF + pTFF3 group and the pTE group. These findings suggested that pTFF3, pEGF, and particularly pTE significantly promote cell migration and contribute to the repair of scratch-induced damage (Figure 6).

### 3.5. Study on the Repairing Effect of Oral Recombinant L. paracasei 27-2 Strains on Intestinal Injury in Colitis Model Mice

On the second day, the mental state of DSS model mice deteriorated. By the third day, they lost weight and had loose stools with occult blood. As time passed, diarrhea worsened, with feces sticking to the anus, visible bloody stools appearing, and body weight continuously dropping. Thus, the DAI score gradually rose. Compared to the DSS modeling group, the pPG/*27-2* group showed a significant decline in diet and activity, along with loose stools and diarrhea. In contrast, the pPG-pTE/*27-2* group improved significantly. They had a normal mental state, and their appetite and activity were barely affected. Some only had loose stools without visible bloody stools. Statistical analysis of DAI scores showed a significant difference (*p* < 0.05) between the pPG/*27-2* and DSS modeling groups, and a highly significant difference (*p* < 0.01) between the pPG-pTE/*27-2* group and the other two. This indicates that oral administration of pPG-pTE/*27-2* can more effectively improve the mice’s condition (Figure 7a).

The H&E staining results showed that the colonic mucosal structure of PBS healthy mice was preserved, displaying typical crypt and glandular structures, many goblet cells, and no pathological damage. In contrast, the colonic epithelial structure of DSS-group mice was severely impaired, with an incomplete structure and absence of crypt formation, resulting in severe pathological damage. Administration of pPG/*27-2* mitigated DSS-induced colon tissue damage and preserved the integrity of most intestinal epithelial structures. Moreover, oral administration of the pPG-pTE/*27-2* mice significantly reduced DSS-induced colon tissue damage, leading to a more robust morphology. The crypt structure was basically restored to its normal state, and a large number of goblet cells were retained (Figure 7b, Appendix A). The MPO value indicates the extent of colonic tissue inflammation in mice. The colon tissues of mice from several experimental groups were treated according to the MPO kit instructions. The colon MPO activity of mice in each treatment group was assessed. Statistical investigation revealed that the MPO value in the DSS modeling group of mice was considerably elevated compared to the normal group (*p* < 0.01). The oral administration of pPG/*27-2* significantly mitigated the elevation of MPO values induced by DSS modeling (*p* < 0.05). The oral treatment of the pPG-pTE/*27-2* group considerably mitigated the elevation of MPO values induced by DSS modeling (*p* < 0.01) and alleviated intestinal inflammation (Figure 7c, Appendix A).

Total RNA was extracted from the colon tissues of mice in each experimental group and homogenized with PBS buffer. Western blot and real-time fluorescence quantitative PCR were used to assess the expression of intestinal epithelial tight-junction proteins Claudin-2, Occludin, and ZO-1. The Western blot analysis indicated that, compared to the PBS-treated group, the expression of tight-junction proteins in the DSS group was markedly reduced (*p* < 0.01). In comparison to both the DSS model group and the PBS group, the expression levels of colonic tight-junction proteins Claudin-2, Occludin, and ZO-1 in the oral pPG-pTE/*27-2* group were substantially elevated (*p* < 0.05, *p* < 0.01) (Figure 7d).

### 3.6. Changes in Colon Length

Colonic shortening is a significant feature linked to colitis. After euthanasia, the entire colon, from the cecum to the anus, of each mouse was meticulously dissected, and its length was accurately measured (Figure 8a). In comparison to the PBS healthy group, the colon length in the DSS modeling group was markedly reduced; however, the oral administration of recombinant *L. paracasei 27-2* strains mitigated the extent of colon shortening to variable degrees. The statistical analysis of colon length indicated that, in comparison to the DSS modeling group, the colon length of mice in the oral pPG/*27-2* group exhibited a significant enhancement (*p* < 0.05). Furthermore, mice in the oral pPG-pTE/*27-2* group demonstrated a significant improvement relative to both the DSS and pPG/*27-2* groups. The length of the colon was markedly reduced (*p* < 0.01) (Figure 8b).

### 3.7. Cytokine Detection Results

Following modeling, the concentrations of serum inflammatory markers in each group were assessed to delineate the inflammatory status. ELISA results demonstrated that the pro-inflammatory cytokines IL-1β, IL-6, and TNF-α and the anti-inflammatory cytokine IL-10 were considerably elevated in DSS model mice. Compared with the DSS group, blood levels of IL-1β, IL-6, and TNF-α in mice from the pPG/*27-2* group and the pPG-pTE/*27-2* group were diminished, although the level of IL-10 was elevated. Significant changes in IL-1β, IL-6, and IL-10 were seen between the two groups; however, no significant difference was noted in level of TNF-α. The findings indicated that the oral pPG-pTE/*27-2* group may modulate inflammatory variables and diminish the inflammatory response induced by DSS modeling (Figure 9).

## 4. Discussion

The gastrointestinal mucosal barrier serves as the body’s primary defense system against external pathogens. Although conventional vaccination strategies have proven effective in preventing systemic infections, their efficacy against mucosal pathogens remains limited due to their inability to elicit strong mucosal immune responses. The mucosal layer, composed of glycoprotein complexes, forms a structural matrix that supports the colonization and adhesion of specific bacterial species, including probiotic strains such as *Lactobacillus* [26]. The utilization of *Lactobacillus* as a delivery platform for expressing therapeutic peptides or protective antigens represents a significant advancement in mucosal vaccination and targeted treatment strategies. The inherent limitations of pTFF3 and pEGF bioavailability following oral administration result in their premature adsorption to small intestinal mucus, leading to rapid clearance before reaching the cecum and ultimately compromising their therapeutic efficacy in the colonic region. This work used *L. paracasei 27-2* (in previous studies, oral administration for seven days resulted in significant colonization in the intestine), isolated from piglet intestines in the laboratory, as the host bacterium to augment the probiotic effectiveness of pTFF3 and pEGF [27].

TFF3 serves as a crucial motility factor that plays an indispensable role in maintaining gastrointestinal mucosal integrity and facilitating epithelial repair following injury. This peptide mediator is particularly important for promoting mucosal healing through its effects on cell migration and restitution processes [27]. Klaas Vandenbroucke and colleagues successfully expressed TFF in lactobacilli, which protected the integrity of the intestinal barrier and effectively alleviated colitis in mice after oral administration. In addition, lactobacilli expressing TFF1 effectively alleviated oral mucosal damage [28]. The study by Wang et al. has shown that TFF3 may enhance cell migration, facilitate the repair of impaired intestinal mucosal barriers, and have a substantial protective impact on intestinal epithelial cells [29]. EGF, functioning as a mitogen, enhances the proliferation and differentiation of intestinal epithelial cells and facilitates the creation of tiny intestinal villi and crypts, which is crucial for intestinal development and damage healing. Studies have shown that oral administration of EGF can increase the gene expression of tight-junction proteins such as ZO-1, Claudin-1, and Occludin, thereby enhancing the intestinal barrier function of early weaned piglets [7]. In the scratch test, pTFF3, pEGF, and pTE were tested at their optimal concentrations to promote cell proliferation (pPG: 100 ng/mL (total protein concentration); pEGF: 50 ng/mL (2.27 × 10^−2^ µg/mg); pTFF3: 100 ng/mL (4.54 × 10^−2^ µg/mg); pTE: 25 ng/mL (1.14 × 10^−2^ µg/mg); pEGF + pTFF3: pEGF—50 ng/mL (2.27 × 10^−2^ µg/mg), pTFF3—100 ng/mL (4.54 × 10^−2^ µg/mg)). Prior research has shown that TFF3 and EGF have synergistic effects in the healing and rebuilding of injured intestinal mucosa. This research demonstrated that the inclusion of pTFF3, pEGF, and pTE significantly enhanced cell migration and expedited wound healing relative to the control group in cell scratch experiments. The fusion expression of pTE produced a greater area of scratch repair and a more rapid healing rate compared to pTFF3 or pEGF individually. The data indicated that pTE may have a synergistic impact on enhancing cell migration and tissue healing, as shown by this experiment. Furthermore, the co-expression of pTE and pTFF3 proteins did not influence their biological activities. In vitro investigations have shown that the addition of pTFF3, pEGF, and pTE proteins markedly increases the proliferation of IPEC-J2 cells. In summary, in comparison with pPG/*27-2*, pPG-pEGF/*27-2*, and pPG-pTFF3/*27-2*, pPG-pTE/*27-2* manifests a much more outstanding efficacy in facilitating cell migration and proliferation.

DSS-induced colitis models are widely recognized in the scientific community for their exceptional reproducibility and experimental tractability, establishing them as a gold standard for investigating the pathogenesis and therapeutic interventions of intestinal inflammatory disorders in preclinical research [30]. This study demonstrated a significant increase in the disease activity index (DAI), incorporating both diarrhea severity and fecal occult blood scores, in DSS-treated mice compared to PBS-treated healthy controls. Histopathological analysis revealed substantial preservation of colonic mucosal architecture in control animals, in stark contrast to the extensive epithelial damage observed in DSS-induced colitis models. These collective findings validate the successful establishment of a reproducible and physiologically relevant colitis model. This study revealed that the disease activity index (comprising diarrhea score and fecal occult blood index) in DSS-treated mice was significantly elevated compared to that of PBS healthy controls, which exhibited a more intact colonic structure than the DSS group in vivo. These findings confirmed the successful establishment of the model [31]. Moreover, while the DSS model group and the oral pPG/*27-2* group exhibited comparable results, the treatment of oral pPG-pTE/*27-2* markedly reduced colon shortening. Furthermore, MPO activity, used to assess neutrophil recruitment [32], was significantly increased in DSS-treated animals relative to those administered PBS. This indicates that the oral treatment of expressing pPG-pTE/*27-2* may reduce neutrophil infiltration and inflammation in these animals. The characterized tight-junction proteins, such as Claudin-2, Occludin, and ZO-1, act in concert to uphold the integrity of the intestinal epithelial architecture [33]. This research revealed that the expression levels of tight-junction proteins Claudin-2, Occludin, and ZO-1 were markedly reduced in the colon tissues of DSS model mice. Compared to the DSS model group and the group treated with pPG/*27-2*, there was a notable elevation in the expression levels of tight-junction proteins in the intestinal mucosa after treatment with oral pPG-pTE/*27-2*.

Evaluating secreted cytokines is essential for determining the efficacy of a genetically engineered vaccine [34]. Tumor Necrosis Factor-alpha (TNF-α) represents a central mediator in the inflammatory cascade of enteritis, playing a crucial role in disease pathogenesis through its pro-inflammatory actions. Mechanistically, TNF-α exerts its biological effects by activating the Nuclear Factor-kappa B (NF-κB) signaling pathway, which subsequently regulates the expression of various inflammatory mediators and immune responses [35]. Subsequently, the activated NF-κB triggers the synthesis of additional pro-inflammatory cytokines, IL-1β and IL-6 [36]. Conversely, the anti-inflammatory cytokine IL-10 may suppress major histocompatibility complex (MHC) class II antigens and co-stimulatory molecules, therefore diminishing the production of pro-inflammatory cytokines and chemokines [37]. This investigation revealed statistically significant alterations in IL-1β, IL-6, TNF-α, and IL-10, demonstrating that DSS modeling may markedly increase the blood levels of these inflammatory cytokines in mice. LAB, being probiotics with resistance to acid and bile salts, as well as adhesion properties, may colonize the intestines of animals. Consequently, their modulation of intestinal mucosal barrier function and immunological function in animals is very significant [38]. Moreover, recombinant *Lactobacillus* may exert a regulatory effect on the transcriptional expression of cytokines, specifically IL-10 and transforming growth factor-β (TGF-β), pattern-recognition receptors, namely Toll-like receptor 2 (TLR-2) and Toll-like receptor 4 (TLR-4), as well as genes associated with major inflammatory regulatory factors, such as NF-κB and single immunoglobulin Interleukin-1 receptor-related molecule. This regulatory action ultimately leads to the attenuation of inflammatory responses [39]. This research demonstrated that oral delivery of pPG/*27-2* ameliorated DSS-induced colitis in mice. In contrast to the DSS model and the oral pPG/*27-2* group, the oral pPG-pTE/*27-2* group exhibited dramatically decreased levels of the pro-inflammatory cytokines IL-1β, IL-6, and TNF-α. Concurrently, the expression level of the anti-inflammatory cytokine IL-10 was markedly elevated. This suggests that the oral treatment of pPG-pTE/*27-2* may effectively modulate inflammatory factors and mitigate intestinal inflammation resulting from DSS-induced colitis. The data suggest that *L. paracasei 27-2* synergizes with pTFF3 and pEGF, and their combination administration may enhance treatment outcomes more effectively than their individual use.

## 5. Conclusions

In summary, the oral administration of pPG-pTE/*27-2* has been demonstrated to alleviate DSS-induced colitis in mice. This treatment not only suppresses the inflammatory response and mitigates intestinal damage but also enhances the mucosal barrier function of intestinal epithelial cells by upregulating the expression of tight-junction proteins. This research offers valuable insights into the treatment of IBD, thereby laying a solid foundation for further exploration in this field.

## 6. Research Limitations

Selecting an appropriate model according to research requirements is of utmost importance, given that distinct models can yield diverse phenotypes. In this experiment, only BALB/c mice were employed, which presents certain limitations. Additionally, the protein concentration range employed in this study was relatively narrow, potentially overlooking effects at lower or higher concentrations. The protein used in this study had a limited purity level, which might have introduced contaminants that interfered with the accurate measurement of its biological activity, leading to potential over- or under-estimation of the protein’s true function. This study marks the initial stage in exploring the characteristics of the target proteins. Nevertheless, we recognize that in vivo expression data are of crucial importance, and future research will be centered around designing and conducting relevant experiments for further exploration.

## Figures and Tables

**Figure 1 vetsci-12-00365-f001:**
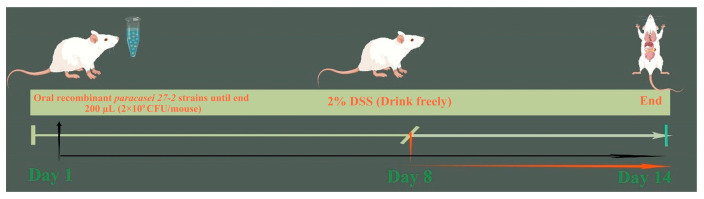
DSS-induced model. Eight mice were included in each group and housed separately in cages. Oral recombinant *L. paracasei 27-2* strains: From day 1 to day 14 of the model establishment period, each mouse in this group was orally administered 200 μL of recombinant *L. paracasei 27-2* strains daily. Meanwhile, from the 8th day to the 14th day, each mouse was allowed to freely drink a 2% DSS solution every day. PBS group: Mice in this group were orally fed 200 μL of PBS daily from day 1 to day 14. DSS group: Each mouse in this group received 200 μL of PBS daily from day 1 to day 7. Subsequently, from the 8th day to the 14th day, each mouse was allowed to freely drink a 2% DSS solution every day. During the model establishment process, the mental state and body weight of each mouse were recorded daily. Additionally, fecal samples were collected to detect fecal occult blood. At the end of the model establishment, all mice were sacrificed.

**Figure 2 vetsci-12-00365-f002:**
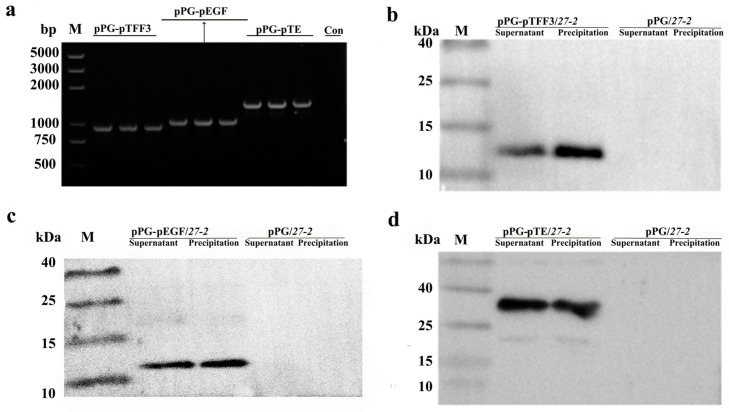
The recombinant *L. paracasei 27-2* strains expressing pTFF3 and pEGF were identified by PCR and Western blot. After culturing the recombinant *L. paracasei 27-2* strains for 16 h, PCR identification was carried out for their plasmids using the primer pair pPG-F/R (**a**). Furthermore, proteins in the culture supernatants and bacterial cell pellets were separately collected. Mouse anti-flag monoclonal antibody was used as the primary antibody, and goat anti-mouse IgG labeled with HRP was used as the secondary antibody for Western blot detection (**b**–**d**).

**Figure 3 vetsci-12-00365-f003:**
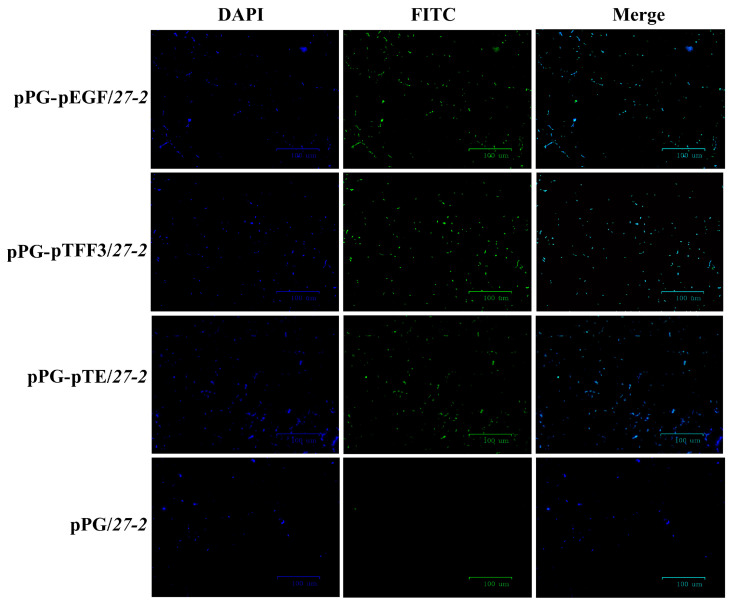
Identification of proteins expressed in recombinant *L. paracasei 27-2* strains by IFA. After culturing the recombinant *L. paracasei 27-2* strains for 16 h, the bacterial cell pellets were separately collected, washed, and then fixed on the glass slides. Mouse anti-flag monoclonal antibody was used as the primary antibody, and goat anti-mouse IgG labeled with FITC was used as the secondary antibody (green). Subsequently, the cell nuclei were stained with DAPI (blue), and the results were observed under an inverted fluorescence microscope.

**Figure 4 vetsci-12-00365-f004:**
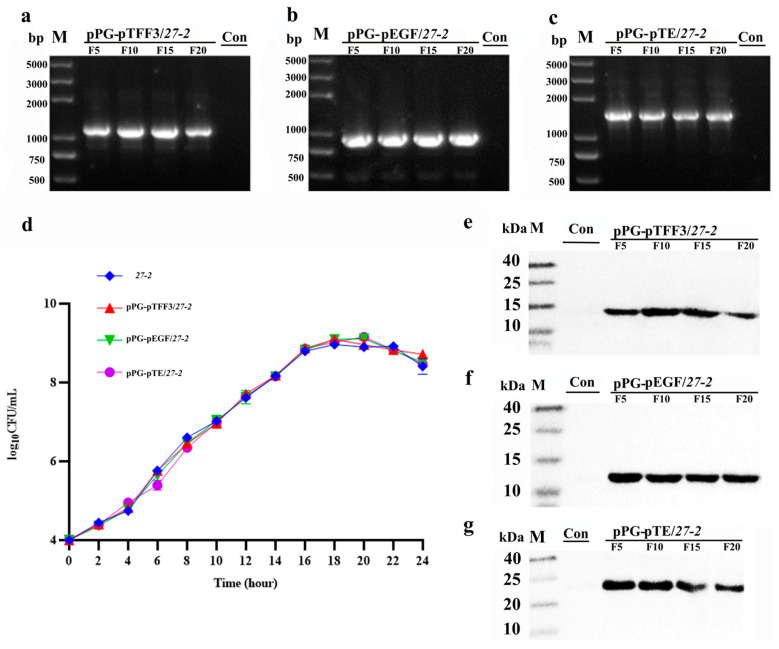
The growth characteristics and stability of recombinant *L. paracasei 27-2* strains were analyzed. The recombinant *L. paracasei 27-2* strains underwent continuous subculturing for 20 generations. Plasmids were extracted at intervals of every five generations, and PCR identification (**a**–**c**) was executed using the primers pPG-F/R. Subsequently, the bacterial proteins were processed. Western blot (**e**–**g**) was then performed, with mouse anti-flag monoclonal antibody serving as the primary antibody and HRP goat anti-mouse IgG as the secondary antibody. The recombinant *L. paracasei 27-2* strains were inoculated at an inoculation ratio of 1%. The cell suspension was collected at two-hour intervals. The viable cell count was carried out by means of the plate counting method (**d**) (*n* = 3, per group).

**Figure 5 vetsci-12-00365-f005:**
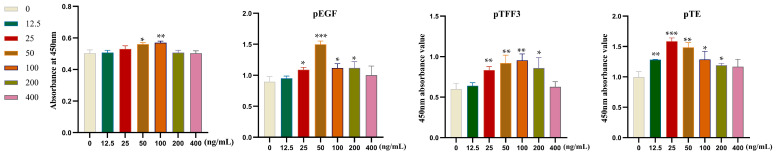
The processed pTFF3, pEGF, and pTE proteins at varying concentrations (0, 12.5 ng/mL (5.68 × 10^−3^ µg/mg), 25 ng/mL (1.14 × 10^−2^ µg/mg), 50 ng/mL (2.27 × 10^−2^ µg/mg), 100 ng/mL (4.54 × 10^−2^ µg/mg), 200 ng/mL (9.08 × 10^−2^ µg/mg), and 400 ng/mL (1.82 × 10^−1^ µg/mg)), with the unit type specified elsewhere in the experimental context, were dispensed into a 96-well plate populated with IPEC-J2 cells that had been pre-cultured for 12 h (100 μL per well). The pPG group represents the total protein concentration and serves as a control for normalization in our protein quantification experiments. Absorbance values at an optical density of 450 nm (OD450) were measured at 30 min intervals. This measurement process was continued until the obtained values entered the optimal reading range defined by the CCK-8 kit protocol. For each protein sample, six replicate wells were set up, and the entire experimental setup was replicated three times in parallel for each group to ensure statistical robustness. Bars represent the mean ± standard error value of each group (* *p* < 0.05, ** 0.01 < *p* < 0.05, *** *p* < 0.01 vs. 0 ng/mL, *n* = 3 per group).

**Figure 6 vetsci-12-00365-f006:**
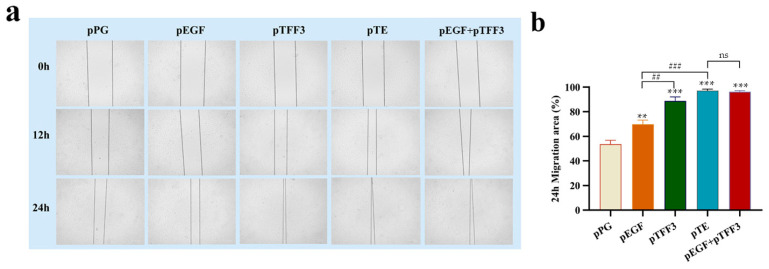
The analysis of the migration effect of the target protein on IPEC-J2. Using the optimal concentrations of each protein for cell proliferation (pPG: 100 ng/mL (total protein concentration); pEGF: 50 ng/mL (2.27 × 10^−2^ µg/mg); pTFF3: 100 ng/mL (4.54 × 10^−2^ µg/mg); pTE: 25 ng/mL (1.14 × 10^−2^ µg/mg); pEGF + pTFF3: pEGF—50 ng/mL (2.27 × 10^−2^ µg/mg), pTFF3—100 ng/mL (4.54 × 10^−2^ µg/mg)), the above-mentioned proteins were added, respectively, after the cell scratch assay. The healing area of the scratch was observed under an inverted fluorescence microscope at 0 h, 12 h, and 24 h (**a**). Bars represent the mean ± standard error value of each group (**b**) (** 0.01 < *p* < 0.05, *** *p* < 0.01 vs. pPG; ^##^ 0.01 < *p* < 0.05, ^###^ *p* < 0.01 vs. pTE; ns represents *p* > 0.05; *n* = 3 per group).

**Figure 7 vetsci-12-00365-f007:**
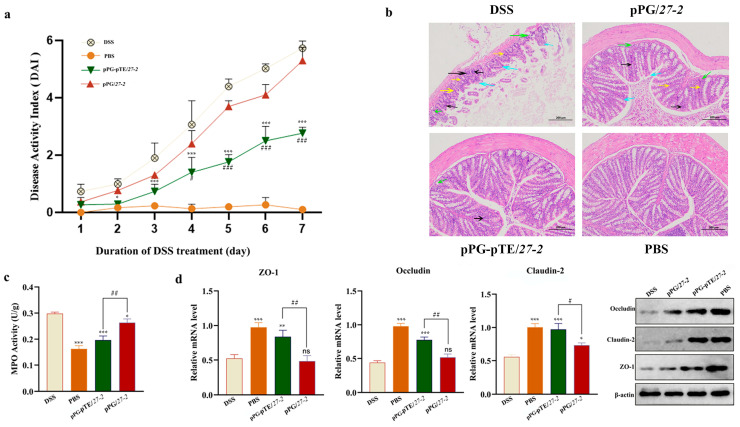
During the period of DSS modeling, each mouse was weighed daily, and its mental state was recorded. The mice were scored according to the DAI scoring table (Table 2) (**a**). After the completion of the modeling, a section of the colon (approximately 1 cm in length) near the rectum was excised from the mice of each group for H&E staining (**b**). The black arrow denotes the infiltration of inflammatory cells, the yellow arrow indicates crypt loss, the green arrow signifies the loss of glandular structure, and the blue arrow represents the loss of goblet cells. Effects of different treatments on colonic MPO activity (**c**). Approximately 0.1 g of the colon from each mouse in each group was separately collected, and the activity of MPO was determined using a myeloperoxidase assay kit. The mRNA transcription levels of tight-junction proteins (ZO-1, Occludin, and Claudin-2) in colon samples (0.1 g) from each mouse group were measured. Data were analyzed via the 2^−△△Ct^ method. For protein analysis, colon tissues (1 g) from each group were used for protein extraction, followed by lysis and quantification. Following the normalization of protein quantities across all experimental groups, a Western blot assay was conducted. Mouse anti-ZO-1, anti-Occludin, and anti-Claudin-2 monoclonal antibodies served as primary antibodies, with horseradish peroxidase-labeled goat anti-mouse IgG as the secondary antibody (**d**). Bars represent the mean ± standard error value of each group (* *p* < 0.05, ** 0.01 < *p* < 0.05, *** *p* < 0.01 vs. DSS; ^#^ *p* < 0.05, ^##^ 0.01 < *p* < 0.05, ^###^ *p* < 0.01 vs. pTE/*27-2*; *n* = 3, per group).

**Figure 8 vetsci-12-00365-f008:**
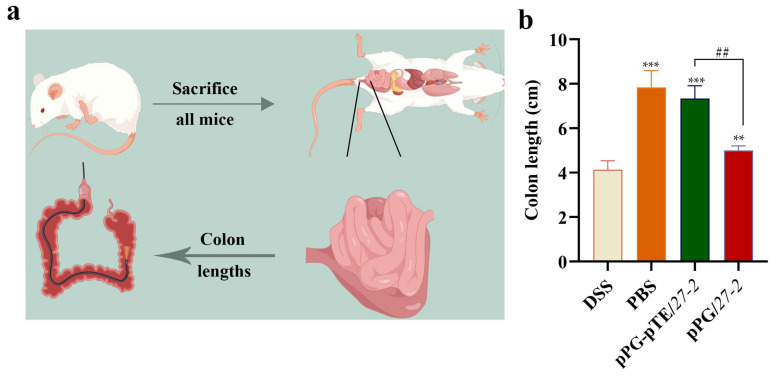
Effects of different treatments on colon length in colitis mice. The length of the colon of each group of mice was measured (**a**). Bars represent the mean ± standard error value of each group (**b**) (** 0.01 < *p* < 0.05, *** *p* < 0.01 vs. DSS; ^##^ 0.01 < *p* < 0.05 vs. pTE/*27-2*; *n* = 3, per group).

**Figure 9 vetsci-12-00365-f009:**
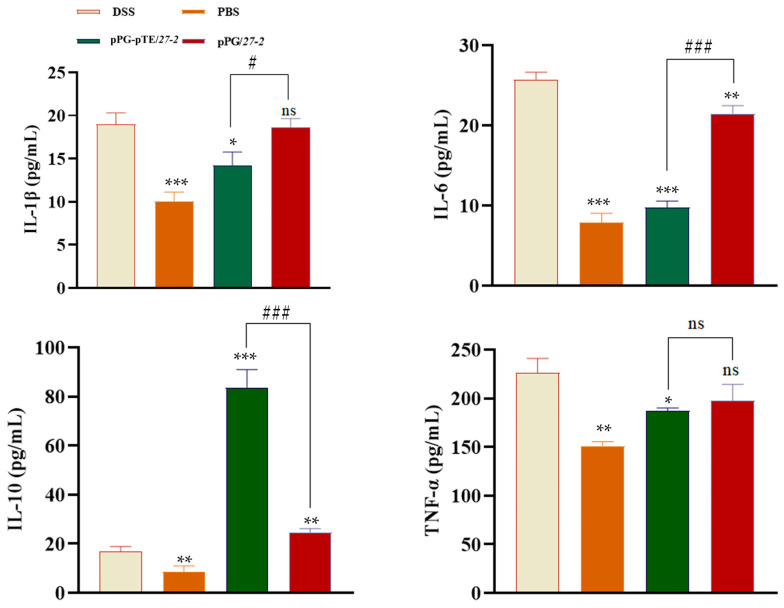
Determination of inflammatory secretion level. The serum samples were collected from each group of mice on the 14th day. The concentrations of pro-inflammatory factors (IL-6, IL-1β, TNF-α) and anti-inflammatory factor (IL-10) were measured using the corresponding ELISA kits. Bars represent the mean ± standard error value of each group (* *p* < 0.05, ** 0.01 < *p* < 0.05, *** *p* < 0.01 vs. DSS; ^#^ *p* < 0.05, ^###^ *p* < 0.01 vs. pTE/*27-2*; ns represents *p* > 0.05; *n* = 3, per group).

**Table 1 vetsci-12-00365-t001:** Experiment design.

Group	Oral Dosage, Per Mouse Per Day	Number
PBS	200 μL (Days 1 to 14)	8
pPG-pTE/*27-2*	200 μL (2 × 10^9^ CFU, days 1 to 14, 2%DSS (free access to water, days 8 to 14))	8
pPG/*27-2*	200 μL (2 × 10^9^ CFU, days 1 to 14, 2%DSS (free access to water, days 8 to 14))	8
DSS	2%DSS (free access to water, days 8 to 14)	8

Note: Meanwhile, from day 8 to day 14, each mouse in each group except for the PBS group was free to drink 2% DSS solution daily. Each mouse in the DSS group drank 200 μL of PBS per day for 1 to 7 days. Under ideal conditions, approximately 0.4 µg of exogenous protein was expressed at this specific dose.

**Table 2 vetsci-12-00365-t002:** Disease activity index.

Weight Loss (%)	Occult Blood/Bloody	Fecal Property	Score
0	Occult blood negative	Normal	0
1–5	Occult blood negative	Loose	1
6–10	Positive for occult blood	Loose	2
11–15	Positive for occult blood	Loose stools	3
>15	Naked-eye bloody stool	Loose stools	4

**Table 3 vetsci-12-00365-t003:** Identification of pTFF3, pEGF, and pTE expressed in supernatant of the recombinant *L. paracasei 27-2* strains.

Incubation Time	Expression in Supernatants of Recombinant *L. paracasei 27-2* Strains pTFF3, pEGF, and pTE (μg/mg)		
	pTFF3	pEGF	pTE
6 h	0.15 ± 0.02	0.10 ± 0.02	0.07 ± 0.03
10 h	0.35 ± 0.01	0.17 ± 0.02	0.19 ± 0.04
14 h	0.45 ± 0.04	0.28 ± 0.01	0.30 ± 0.05
18 h	0.48 ± 0.02 ^a^	0.50 ± 0.01 ^a^	0.37 ± 0.05 ^b^
22 h	0.30 ± 0.02	0.35 ± 0.02	0.31 ± 0.03
24 h	0.20 ± 0.03	0.23 ± 0.01	0.25 ± 0.05

Note: The culture supernatants of the recombinant *L. paracasei 27-2* strains were separately collected at cultivation durations of 6 h, 10 h, 14, 18 h, and 22 h. The data were presented as the mean ± standard deviation (*n* = 3). For different groups, the presence of distinct letters indicated a significant difference (*p* < 0.05), while the same letter indicated no significant difference between groups (*p* > 0.05).

**Table 4 vetsci-12-00365-t004:** Identification of pTFF3, pEGF, and pTE expressed in cell lysates of the recombinant *L. paracasei 27-2* strains.

Incubation Time	Expression in Supernatants of Recombinant *L. paracasei 27-2* Strains pTFF3, pEGF, and pTE (μg/mg)		
	pTFF3	pEGF	pTE
6 h	0.18 ± 0.02	0.16 ± 0.01	0.12 ± 0.01
10 h	0.35 ± 0.01	0.28 ± 0.02	0.27 ± 0.03
14 h	0.45 ± 0.01	0.74 ± 0.06	0.45 ± 0.01
18 h	0.95 ± 0.01 ^a^	0.91 ± 0.07 ^a^	0.90 ± 0.07 ^a^
22 h	0.80 ± 0.01	0.41 ± 0.03	0.51 ± 0.06
24 h	0.57 ± 0.01	0.36 ± 0.01	0.39 ± 0.04

Note: The cell lysates of the recombinant *L. paracasei 27-2* strains were separately collected at cultivation durations of 6 h, 10 h, 14, 18 h, and 22 h. The data were presented as the mean ± standard deviation (*n* = 3). For different groups, the presence of distinct letters indicated a significant difference (*p* < 0.05), while the same letter indicated no significant difference between groups (*p* > 0.05).

## Data Availability

The data that support the findings of this study are openly available in Mendeley Data at http://doi.org/10.17632/kmzz7dsxyt.1.

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
