# Peer review of "Lactobacillus paracasei Expressing Porcine Trefoil Factor 3 and Epidermal Growth Factor: A Novel Approach for Superior Mucosal Repair"

_vetsci, 2025, doi:10.3390/vetsci12040365_

Round 1
Reviewer 1 Report
Comments and Suggestions for Authors
Manuscript vetsci-3519735 present interesting results implying on recombinant Lactobacillus paracasei strains expressing pTFF3 and/or pEGF as potential immunomodulators for control of IBD or delivery platform for oral immunization. However, it is hard to follow up the manuscript in current form and to evaluate data properly. Manuscript suffers from inconsistency in labeling and needs revision by an English lecturer. In addition, legends to the figures are incomplete and Material & Methods section should be improved. Hence, major revision is recommended.
Besides grammar issue, I think that misusage of some terms occurred as well.
Major concern:
- Reference 2 is not appropriate for citing at that point… this is a review on intestinal growth factors in IBD, but not on Lactobacillus as a vehicle
- Introduction, 4th paragraph – sentence “To enhance the probiotic properties of TFF3 and EGF, as well as their capacity to repair intestinal mucosal damage, this study employed Lactobacillus paracasei, a bacterium capable of colonizing the intestine, as the host to co-express TFF3 and epidermal growth factor” has to be revised. This comment also refers to “This work used Lactobacillus 27-2, isolated from piglet intestines in the laboratory…” (Discussion, 1st paragraph). Probiotic properties could be assigned to live microorganism, in those specific cases to paracasei
- There are some mess with Lactopacillus (genus) vs Lactobacillus paracasei (species) due to inappropriate citing in the manuscript. After first mentioning, Lactobacillus paracasei shoud be written as paracasei (followed by strain 27-2, pPG-SP-pTFF3/27-2, pPG-SP-pEGF/27-2 or pPG-SP-pTE/27-2) throughout the entire manuscript. Please check and correct.
- Section 2.2 – it has to be clearly pointed that pTFF3, pEGF and pTE are expressed with flag-tag that enable their detection with anti-flag antibodies.
- Section 2.3 - Follow up of pTE expression has to be described as well (via pTFF3 or pEGF or both pTFF3 and pEGF or some other way…). In addition, were there any adjustment of samples prior ELISA testing (for example same concentration of total proteins or same number of bacteria for preparation of cell lysates)? That information is important for proper discussing of the results.
- Section 2.5 – The following has to be addressed:
- Citing of the article by Wingfield P is not sufficient / appropriate. Precipitation by (NH4)2SO4 has to be described briefly, including specific saturation of (NH4)2SO4 solution used for precipitation of the proteins.
- Which concentration was measured – total proteins or specifically pTTF3/pEGF/pTE? Precise information has to be provided (1st paragraph).
- Were proteins controlled for purity by electrophoresis and/or for endotoxin content upon completion of purification? Data should be provided.
- Concentration range used for stimulation and scratch assay has to be provided. Duration of stimulation / incubation has to be indicated as well, including timepoint when measurement started.
- Section 6, including title, Figure 1 and Table 1, has to be revised. Title - I find term “immunization” inappropriate in the context of Section 2.6 (described treatments aim some kind of immunomodulation instead of the induction of antigen-specific immune response). Table 1 – based on the written in Table 1, mice that received pPG-SP-pTE/27-2 or pPG /27-2 treatment were not subjected to disease induction (???) as depicted (it seems) at Figure 1. Figure 1 -Appropriate legend should be provided, making it understandable without referring to the text. In addition, DSS treatment could be indicated by syringe only (without needle… syringe with needle suggests on injection rather than oral treatment). Following details have to be provided in the text:
- When oral feeding with transformed L. paracasei started (with respect to DSS treatment) and what was feeding regime (daily?, dose,…)?
- Housing system – group or individual cage-based housing? In line, monitoring of stool for consistency and blood traces has to be described.
- Section 2.7 – Which part of the colon (location of 1 cm long specimen) were analyzed? What “differentiation” assumes?
- Section 2.8 – Composition of homogenization medium has to be provided. MPO activity was assessed for each colon or pool(s) per group were made?
- Details on statistical analysis has to be included in Material & Methods.
- Appropriate legends for Figures have to be provided, making it understandable without referring to the text. The following details have to be included: description of each plot / picture, description of samples (collection time, …), number of replicates i.e tested samples, way of presentation (mean ± SD, representative…), statistical method (with clearly indicated referent group / samples… referent groups indicating in legends to the Figures 6-7 do not correspond to the plots), explanation for abbreviations
- Section 3.3 – What was load of proteins other than pTFF3/pEGF/pTE? How authors explain bell-shaped relationship between proliferation and stimulatory concentration? Is there possibility for some cytotoxic effect at higher concentrations? Whether cytotoxicity is evaluated?
- Section 3.4 – Result on pTFF3+pEGF stimulation of cell migration is presented at Figure 6b. However, corresponding pictures are not provided at Figure 6a. Besides, there is no mentioning of that stimulation in the text. Comparison of the outcomes of pTE and pTFF3+pEGF stimulations could be valuable as implication on structural and functional characteristics of pTE.
- Section 3.5 – Section should be rewritten (in line with suggestion No. 7 and general suggestion on consistent labelling). In addition, labelling of x-axis at Figure 7a should be changed (there were no infection… for example „Duration of DSS treatment (day)“). In addition, the follow up of the Manuscript would be much easier with consistent presentation of the results (same color for one group and same order of the groups at Figures 7-9)
- Section 3.7 – IL-10 is anti-inflammatory cytokine while TNFa is pro-inflammatory. Revise, please.
- Discussion should be improved:
- Advantages of paracasei pPG-SP-pTE/27-2 over L. paracasei (pPG/27-2) / L. paracasei pPG-SP-pTFF3/27-2 / L. paracasei pPG-SP-pEGF/27-2 should be discussed.
- Limitation of the study should be clearly addressed (for example only BALB/c, purity of the stimulators, evidence on individual pTFF3 and pEGF in vivo effects are not provided…)
- Corresponding references have to be cited in “Klaas Vandenbroucke and colleagues successfully expressed TFF3 in Lactobacillus… pre-serve the integrity of the intestinal barrier, and enhance intestinal health.”
- “Significantly, in comparison to pTFF3 and pEGF, the fusion protein pTE has a more substantial proliferative impact at reduced doses.” Taking into account that proliferation was evaluated by absorbance (Figure 5), this comparison could be made only if all cultures (all stimulators in all doses) were analyzed simultaneously. Please indicate in Section 2.5 if that was a case. Otherwise, sentence should be deleted.
- It has to be clearly indicated which outcomes are common to Lactobacilus and which are species / strain – specific. For example, “… Moreover, recombinant Lactobacillus exerts a regulatory effect…” should be revised as “Moreover, Lactobacillus fermentum (MTCC-5898) exerts a regulatory effect…” or “… Moreover, recombinant Lactobacillus may exert a regulatory effect…”
- List of references has to be check for consistency of citation and accuracy (name of author in low case, spaces, there are some missing of authors / volume / start page No., etc.). Furthermore, I could not find reference 23 (and provided title does not imply that any synergism was studied).
Minor concerns:
- Affiliations – smaller font for a (College of Veterinary Medicine…)
- Introduction, 1st paragraph - Oxazolidinone has to be replaced by oxazolone
- Introduction, 2nd paragraph – abbreviations EGF and TFF have to be introduced at this point and further constantly used through the entire Manuscript. This has to be applied for all abbreviations in the manuscript –check, please!!!
- In vitro / vivo have to be written in Italic
- Introduction, 4th paragraph – trihotropic factor? Please check
- Numerical value and unit have to be separated by space (for example 37°C has to be 37 °C).
- Section 2.3 – working dilution has to be associated with specific antibody (i.e. manufacturer and catalog no. have to be provided for each antibody) as that relationship is not universal. In addition, composition of the diluent solution has to be provided (diluent?, % skim milk in…?)
- Section 2.3, 2nd paragraph – “resistance” and “Cmr” have to be deleted
- Section 2.4 – … for assessing “expression stability” instead of “genetic stability”
- Material & Methods – it has to be written in past tense instead of imperative (sections 2.4, 2.8, 2.9, 2.10)
- Section 2.5, 2nd line – a dot (.) has to be deleted
- Section 2.9 – “…Western and IP lysis buffer containing 1% PMSF” has to be reformulated. Buffer for Western blot containing 1 % proteases inhibitors (PMSF)? Composition of the solution has to be provided.

It is hard to follow up the manuscript in current form. It needs revision by an English lecturer. Besides grammar issue, I think that misuse of some terms occurred as well (for example "genetically modified vaccination").
Author Response
Dear reviewer,
Thank you very much for having our manuscript entitled “Lactobacillus paracasei Expressing Porcine Trefoil Factor 3 and Epidermal Growth Factor: A Novel Approach for Superior Mucosal Repair” (vetsci-3519735) reviewed in a timely and professional manner and for giving us an opportunity to revise the manuscript. We have revised the manuscript carefully, and used Microsoft Word's built-in track changes function to highlight any changes we made (The content may experience changes when opened in WPS.). We tried best to properly answer the questions and suggestions and made revision in the paper according to the comments (changes in yellow). We hope this new version of article will meet the requirement for publication on Veterinary Sciences.
Sincerely,
Fangjie Yin
Replies to Reviewer #1:
Major concern:
- Reference 2 is not appropriate for citing at that point… this is a review on intestinal growth factors in IBD, but not on Lactobacillus as a vehicle
Response: Thank you very much for your comments. Your suggestions are indeed helpful for us. We have made adjustments to the literature, and your suggestions will be extremely helpful in enhancing the quality of our manuscript. The references we replaced are as follows:A small-scale short-term clinical trial has demonstrated the effectiveness of EGF in treating patients with IBD, with more than 80% of the patients achieving clinical remission [2]. (page 2, lines 59-61)
[2] Krishnan K, Arnone B, Buchman A. Intestinal growth factors: potential use in the treatment of inflammatory bowel disease and their role in mucosal healing. Inflamm Bowel Dis. 2011; 17 (1): 410-22. DOI: 10.1002/ibd.21316
- Introduction, 4th paragraph – sentence “To enhance the probiotic properties of TFF3 and EGF, as well as their capacity to repair intestinal mucosal damage, this study employed Lactobacillus paracasei, a bacterium capable of colonizing the intestine, as the host to co-express TFF3 and epidermal growth factor” has to be revised. This comment also refers to “This work used Lactobacillus 27-2, isolated from piglet intestines in the laboratory…” (Discussion, 1st paragraph). Probiotic properties could be assigned to live microorganism, in those specific cases to paracasei
Response: Thank you very much for your valuable comments.
We agree with your comment for the unclear description in our previous manuscript. In response to your suggestion, we have made corresponding modifications in the manuscript, which are highlighted in yellow (page 3, lines 103-107).
- There are some mess with Lactopacillus (genus) vs Lactobacillus paracasei (species) due to inappropriate citing in the manuscript. After first mentioning, Lactobacillus paracaseishoud be written as paracasei (followed by strain 27-2, pPG-SP-pTFF3/27-2, pPG-SP-pEGF/27-2 or pPG-SP-pTE/27-2) throughout the entire manuscript. Please check and correct.
Response: Thank you very much for your helpful comments.
We have carefully revised the manuscript. The revised parts have been highlighted in yellow for your easy identification (page 3, line 97, line 108, line 121, line 126, page 4, line 152, line 154,line 165, line 171, line 173, line 174, page 5, line 178, page 6, line 220, line 222, line 229, line 230, page 7, line 291, page 8, line 311, line 314, line 317, line 320, line 321, line 326, line 327, page 9 , line 334, line 335, line 341, page 10, line 348, line 350, line 355, line 359, line 362, line 366, page 11, line 370, line 371, page 14, line 490, page 16, line 530, page 17, lines 588 and 600).
- Section 2.2– it has to be clearly pointed that pTFF3, pEGF and pTE are expressed with flag-tag that enable their detection with anti-flag antibodies.
Response: Thank you very much for your comments.
We sincerely hope to facilitate your review. In section 2.2 of the manuscript, we have provided thorough explanations, which have been marked in yellow for your convenience (page 3, lines 122-124).
- Section 2.3 - Follow up of pTE expression has to be described as well (via pTFF3 or pEGF or both pTFF3 and pEGF or some other way…). In addition, were there any adjustment of samples prior ELISA testing (for example same concentration of total proteins or same number of bacteria for preparation of cell lysates)? That information is important for proper discussing of the results.
Response: Thank you very much for your helpful comments and those comments are all valuable and very helpful for revising and improving our paper. We are sorry about that we did not describe the procedures clearly.
1) In this research, two distinct ELISA kits were employed to conduct the quantitative detection of the fusion protein. Subsequently, the two sets of resulting data were combined. We have supplemented it in the revised manuscript, on page four, lines 160-161, and marked it in yellow. (n = 3)
2) The procedures for recombinant paracasei 27-2 strains prior to performing the protein quantification ELISA in this study are as follows.
In this study, the recombinant paracasei 27-2 strains that had been activated by streaking were inoculated into MRS broth with chloramphenicol (10 µg ml⁻¹) respectively. When the optical density at 600 nm (OD600) of each group of recombinant paracasei 27-2 strains reached 1, they were introduced into MRS medium at a 1:100 dilution to evaluate the expression levels of the target protein during further culture. Samples were collected at 6, 10, 14, 18, 22, and 24-hour intervals to measure the quantity of target proteins in both the supernatant and bacterial lysate at these time points using commercial pTFF3 and pEGF ELISA kit.
Additionally, supplementary explanations have been provided in the 3rd paragraph of Section 2.3 of the manuscript, which is highlighted in yellow (page 4, lines 152-156).
- Section 2.5 – The following has to be addressed:
- Citing of the article by Wingfield P is not sufficient / appropriate. Precipitation by (NH4)2SO4 has to be described briefly, including specific saturation of (NH4)2SO4 solution used for precipitation of the proteins.
Response: Thank you very much for your suggestions.
This comment is valuable and very helpful for improving our paper. We have made a replacement of this reference. Furthermore, in Section 2.5, additional explanations on the process of treating proteins with ammonium sulfate were provided by us and are presented in yellow for easy identification [1] (Section 2.5, page 5, lines 180-186).
[1] Protein Precipitation Using Ammonium Sulfate. Curr Protoc Protein Sci. 2016;84: A. 3F. 1-A. 3F. 9. DOI: 10.1002/0471140864.psa03fs84
7.- Which concentration was measured – total proteins or specifically pTTF3/pEGF/pTE? Precise information has to be provided (1st paragraph).
Response: Thank you very much for your suggestions.
We did not clearly describe the procedure in this study. As suggested in your revised manuscript, we have added relevant descriptions (page5, line 187-189) about detecting protein expression levels and highlighted them in yellow as follows: “Subsequently, the protein concentration in the supernatant was measured using pTFF3 and pEGF ELISA kits. (Wuhan Enzyme Immunoassay Biotechnology Co., Ltd, Wuhan, China).”
8.-Were proteins controlled for purity by electrophoresis and/or for endotoxin content upon completion of purification? Data should be provided.
Response: Thank you very much for your suggestions.
In our in-vitro experiments, we indeed do not involve protein purification. In our study, we employed ammonium sulfate precipitation solely for protein concentration (not purification) to facilitate detection and accomplish our experimental aims. This approach was deemed most suitable given the unique characteristics of our biological system.
Moreover, considering our experimental technique and method, where Lactobacillus casei acts as an oral delivery carrier, and it does not function in the form of a single protein in applications, the process of protein purification is not essential. In fact, using crude protein extract has proven sufficient for our research purposes.
We truly hope that this explanation adequately addresses your concerns.
9.-Concentration range used for stimulation and scratch assay has to be provided. Duration of stimulation / incubation has to be indicated as well, including timepoint when measurement started.
Response: Thank you very much for your comments.
Your suggestions are indeed helpful to us. We sincerely apologize for the omission of specifying the sample concentration and stimulation time in the cell scratch experiment. We have now thoroughly revised and supplemented the relevant text to provide clear details in the revised manuscript. The revised part is highlighted in yellow in Section 2.5, the third paragraph, lines 205-209.
- -Section 6, including title, Figure 1 and Table 1, has to be revised. Title - I find term “immunization” inappropriate in the context of Section 2.6 (described treatments aim some kind of immunomodulation instead of the induction of antigen-specific immune response). Table 1 – based on the written in Table 1, mice that received pPG-SP-pTE/27-2 or pPG /27-2 treatment were not subjected to disease induction (???) as depicted (it seems) at Figure 1. Figure 1 -Appropriate legend should be provided, making it understandable without referring to the text. In addition, DSS treatment could be indicated by syringe only (without needle… syringe with needle suggests on injection rather than oral treatment). Following details have to be provided in the text:
- When oral feeding with transformed L. paracasei started (with respect to DSS treatment) and what was feeding regime (daily? dose,…)?
- Housing system – group or individual cage-based housing? In line, monitoring of stool for consistency and blood traces has to be described.
Response: Thank you very much for your valuable comments.
Your comments have been invaluable in enhancing the quality of our manuscript. In light of the insufficient expression in Figure 1 and Table 2, we have made comprehensive modifications and supplements in the revised version. In addition, to ensure the consistency of the names of the recombinant paracasei 27-2 strains throughout the text, the recombinant paracasei 27-2 strains were named pPG-pTE/27-2, pPG-pEGF/27-2, pPG-pTFF3/27-2. The updated figure, tables, along with their corresponding captions, are presented as follows:
Figure 1. DSS induced model. Eight mice were included in each group and housed separately in cages. Oral Recombinant paracasei 27-2 strains: From day 1 to day 14 of the model establishment period, each mouse in this group was orally administered 200 μL of recombinant paracasei 27-2 strains daily. Meanwhile, from the 8th day to the 14th day, each mouse was allowed to freely drink a 2% DSS solution every day. PBS Group: Mice in this group were orally fed 200 μL of PBS daily from day 1 to day 14. DSS Group: Each mouse in this group received 200 μL of PBS daily from day 1 to day 7. Subsequently, from the 8th day to the 14th day, each mouse was allowed to freely drink a 2% DSS solution every day. During the model establishment process, the mental state and body weight of each mouse were recorded daily. Additionally, fecal samples were collected to detect fecal occult blood. At the end of the model establishment, all mice were sacrificed.
Table 1. Experiment design.
Group |
Oral dosage, per mouse per day |
Number |
PBS |
200 μL (Days 1 to 14) |
8 |
pPG-pTE/27-2 |
200 μL (2×109 CFU, days 1 to 14) |
8 |
pPG/27-2 |
200 μL (2×109 CFU, days 1 to 14) |
8 |
DSS |
2%DSS (Free access to water, days 8 to 14) |
8 |
Note: Meanwhile, from day 8 to day 14, each mouse in each group except for the PBS group was free to drink 2% DSS solution daily. Each mouse in the DSS group drank 200 μL of PBS per day for 1 to 7 days.
Section 2.6. Animal Model. Page 5, line 210.
Oral dosage, per mouse per day. Pages 5-6, lines 219-224.
Housing system: Eight mice were included in each group and housed separately in cages (page 5, line 212).
Monitoring of stool for consistency and blood traces: Additionally, fecal samples were collected to detect fecal occult blood (page 6, lines 237-237)
- Section 2.7 – Which part of the colon (location of 1 cm long specimen) were analyzed? What “differentiation” assumes?
Response: Thank you very much for the helpful suggestions.
To rectify this, we have provided a more detailed clarification within Section 2.7 of the manuscript (page 7, lines 254-256), which has been conspicuously highlighted in yellow for easy identification. Furthermore, we regret the errors that were present in the manuscript. Specifically, the incorrect term 'differentiation' has been thoroughly removed from the newly revised version. We are committed to presenting a more accurate and polished manuscript, and we appreciate your understanding and patience.
- Section 2.8 – Composition of homogenization medium has to be provided. MPO activity was assessed for each colon or pool(s) per group were made?
Response: Thank you very much for the valuable suggestions.
We have changed the corresponding position in the manuscript to “Tissue was weighed', And a 5% homogeneity (1:19 w/v in PBS) was prepared” and highlighted in yellow. (Section 2.8, page 7, lines 267-268)
MPO levels were determined using an MPO ELISA kit. The detailed procedure is described in Section 2.8. First, the tissue was weighed, and a 5% homogenate was prepared by adding the tissue to PBS at a weight/volume ratio of 1:19. Then, the MPO activity in the colon samples was measured using an ELISA kit (Nanjing Jiancheng Biotechnology Co., Ltd, Nanjing, China) following the manufacturer's instructions. The results were expressed as MPO activity units per gram of tissue, calculated based on the absorbance at 450 nm.
- Details on statistical analysis has to be included in Material & Methods.
Response: Thank you very much for the valuable suggestions.
Your suggestions have been of immeasurable value in enhancing the quality of our manuscript. We sincerely appreciate your insights. In response, we have incorporated a new section into the manuscript. Moreover, we have furnished additional detailed explanations, which are prominently highlighted in yellow within Section 2.11 of the revised manuscript (page 8).
“2.11. Statistical Analysis”
All the experiments in this study were repeated three times. The data were statistically analyzed using the Two-way ANOVA method with the Graphpad Prism 8.0 software. As shown in the significant difference analysis of P-values: * P < 0.05, 0.01 < **P < 0.05 and ***P < 0.01, while ns represents P > 0.05, and # represents the same significance range as *.
Values indicate mean ± standard deviation (n = 3); bolded font indicates maximum values; different letters indicate significant differences between groups (P < 0.05); the same letters indicate non-significant differences (P > 0.05).
- Appropriate legends for Figures have to be provided, making it understandable without referring to the text. The following details have to be included: description of each plot / picture, description of samples (collection time, …), number of replicates i.e tested samples, way of presentation (mean ± SD, representative…), statistical method (with clearly indicated referent group / samples… referent groups indicating in legends to the Figures 6-7 do not correspond to the plots), explanation for abbreviations
Response: Thank you very much for your comments.
We apologize for our mistakes and have replaced Figures 6-7 with more intuitive figures in the new manuscript. In addition, we have added more detailed legends below each figure. These legends not only explain the displayed content, but also guide readers to understand the key points. We believe that these improvements will greatly enhance the readability of our manuscript, enabling reviewers and future readers to better grasp the content of our research. The changed Figures are shown below.
Figure 7. During the period of DSS modeling, each mouse was weighed daily, and its mental state was recorded. The mice were scored according to the DAI scoring table (Table 2) (a). After the completion of the modeling, a section of the colon (approximately 1 cm in length) near the rectum was excised from the mice of each group for H&E staining (b). The black arrow denotes the infiltration of inflammatory cells, the yellow arrow indicates crypt loss, the green arrow signifies the loss of glandular structure, and the blue arrow represents the loss of goblet cells. Effects of different treatments on colonic MPO activity (c). Approximately 0.1 g of the colon from each mouse in each group was separately collected, and the activity of MPO was determined using a myeloperoxidase assay kit. The mRNA transcription levels of tight-junction proteins (ZO-1, Occludin, Claudin-2) in colon samples (0.1 g) from each mouse group were measured. Data were analyzed via the 2-△△Ct method. For protein analysis, colon tissues (1 g) from each group were used for protein extraction, followed by lysis and quantification. Following the normalization of protein quantities across all experimental groups, a Western blot assay was conducted. Mouse anti-ZO-1, anti-Occludin, and anti-Claudin-2 monoclonal antibodies served as primary antibodies, with horseradish peroxidase-labeled goat anti-mouse IgG as the secondary antibody (d). Bars represent the mean ± standard error value of each group (*P < 0.05, **0.01 < P < 0.05, ***P < 0.01 vs DSS, #P < 0.05, ##0.01 < P < 0.05, ###P < 0.01 vs pTE/27-2, n = 3, per group).
Figure 8. Effects of different treatments on colon length in colitis mice. Measure the length of the colon of each group of mice. Bars represent the mean ± standard error value of each group (*P < 0.05, **0.01 < P < 0.05, ***P < 0.01 vs DSS, #P < 0.05, ##0.01 < P < 0.05, ###P < 0.01 vs pTE/27-2, n=3, per group).
- Section 3.3 – What was load of proteins other than pTFF3/pEGF/pTE? How authors explain bell-shaped relationship between proliferation and stimulatory concentration? Is there possibility for some cytotoxic effect at higher concentrations? Whether cytotoxicity is evaluated?
Response: Thank you very much for the helpful suggestions.
1) We sincerely appreciate your question. In the course of this experiment, no proteins other than the pTE protein were detected. We employed ammonium sulfate precipitation solely for protein concentration (not purification) to facilitate detection and accomplish our experimental aims. This approach was deemed most suitable given the unique characteristics of our biological system. In future studies, we will make earnest efforts to conduct in - depth research on their detection.
- When the stimulus concentration is low, the number of receptors on the cell surface that bind to the stimulus is limited, and the activated intracellular signaling pathways are relatively weak. With the increase of stimulation concentration, more receptors are activated, and the signaling pathway gradually strengthens, thereby promoting cell proliferation, and the proliferation rate shows an upward trend. Based on this characteristic, the proliferation effect of these factors on cells can be weakened by inhibiting their receptors[1-3]. When the stimulus concentration reaches a certain level, signaling molecules are in a state of maximum activation. At this point, continuing to increase the stimulation concentration may lead to interruption or overactivation of signaling pathways, triggering intracellular stress responses and inhibiting cell proliferation[4-6]. Additional explanations have also been provided in section 3.3 of the manuscript, highlighted in yellow (page 11, lines 385-395).
- The CCK-8 experiment was used to evaluate the effects of two proteins (pTFF3, pEGF) on cell proliferation, which to some extent indicates the cytotoxic effects of the two proteins on Intendal Porcine Epithelial Cell Line J2 within this concentration range (0, 12.5, 25, 50, 100, 200 ng/mL).
References:
- Planchard D, Jänne PA, Cheng Y. Osimertinib with or without Chemotherapy in EGFR -Mutated Advanced NSCLC. N Engl J Med. 2023; 389 (21): 1935-1948. DOI: 1056/NEJMoa2306434
- Li S, Zhang H, Ning T. MiR-520b/e Regulates Proliferation and Migration by Simultaneously Targeting EGFR in Gastric Cancer. Cell Physiol Biochem. 2016; 40 (6): 1303-1315. DOI: 1159/000453183
- Yu HA, Goto Y, Hayashi H. HERTHENA-Lung01, a Phase II Trial of Patritumab Deruxtecan (HER3-DXd) in Epidermal Growth Factor Receptor-Mutated Non-Small-Cell Lung Cancer After Epidermal Growth Factor Receptor Tyrosine Kinase Inhibitor Therapy and Platinum-Based Chemotherapy. J Clin Oncol. 2023; 41 (35): 5363-5375. DOI: 1200/JCO.23.01476
- Zhao X, Dai W, Zhu H. Epidermal growth factor (EGF) induces apoptosis in a transfected cell line expressing EGF receptor on its membrane [J]. Cell Biology International, 2013, 30 (8): 653-658. DOI: 1016/j.cellbi.2006.04.004.
- Zhang Y, Liu Y, Wang L. The expression and role of trefoil factors in human tumors. Transl Cancer Res. 2019; 8 (4): 1609-1617. DOI: 21037/tcr.2019.07.48
- Sagmeister T, Gubensäk N, Buhlheller C. The molecular architecture of Lactobacillus S-layer: Assembly and attachment to teichoic acids. Proc Natl Acad Sci U S A. 2024; 121 (24): DOI: 10.1073/pnas.2401686121
- Section 3.4 – Result on pTFF3+pEGF stimulation of cell migration is presented at Figure 6b. However, corresponding pictures are not provided at Figure 6a. Besides, there is no mentioning of that stimulation in the text. Comparison of the outcomes of pTE and pTFF3+pEGF stimulations could be valuable as implication on structural and functional characteristics of pTE.
1) Response: Thank you for your comments. We apologize for our mistakes and have changed in the new manuscript (section 3.4).
Figure 6. The analysis of the migration effect about the target protein on IPEC-J2. Using the optimal concentrations of each protein for cell proliferation (pEGF: 50 ng/mL, pTFF3: 100 ng/mL, pTE: 25 ng/mL, pEGF+pTFF3: pEGF-50 ng/mL, pTFF3-100 ng/mL), the above-mentioned proteins were added respectively after the cell scratch assay. The healing area of the scratch was observed under an inverted fluorescence microscope at 0 h, 12 h, and 24 h. Bars represent the mean ± standard error value of each group (*P < 0.05, **0.01 < P < 0.05, ***P <0.01 vs pPG, #P < 0.05, ##0.01 < P < 0.05, ###P < 0.01 vs pTE, ns represents P>0.05, n=3 per group).
2) The objective of this experimental group was to explore whether the co-expression of pTFF3 and pEGF impacts their functions. The results indicated that the concurrent addition of these two proteins exerted a significant effect in comparison to the control group. Nevertheless, the pTE group demonstrated more pronounced cell-promoting effects. As a result, we refrained from using pTFF3 and pEGF in combination. The relevant discussion has been incorporated (page 12, line 412, lines 413-416 page 16, lines 549-551)
- Section 3.5 – Section should be rewritten (in line with suggestion No. 7 and general suggestion on consistent labelling). In addition, labelling of x-axis at Figure 7a should be changed (there were no infection… for example „Duration of DSS treatment (day) “). In addition, the follow up of the Manuscript would be much easier with consistent presentation of the results (same color for one group and same order of the groups at Figures 7-9)
Response: Thank you for your comments.
- We have made revisions in section 3.5 of the new manuscript and throughout entire manuscript.
- The label on the x-axis in Figure 7a has been revised to "Duration of DSS treatment (day)", and the modified Figure 7 is presented below.
3)The suggestions regarding the grouping colors and arrangement order in Figures 7-9 have been implemented as shown below, and the revised figures have been inserted into the new manuscript.
Figure 7. During the period of DSS modeling, each mouse was weighed daily, and its mental state was recorded. The mice were scored according to the DAI scoring table (Table 2) (a). After the completion of the modeling, a section of the colon (approximately 1 cm in length) near the rectum was excised from the mice of each group for H&E staining (b). The black arrow denotes the infiltration of inflammatory cells, the yellow arrow indicates crypt loss, the green arrow signifies the loss of glandular structure, and the blue arrow represents the loss of goblet cells. Effects of different treatments on colonic MPO activity (c). Approximately 0.1 g of the colon from each mouse in each group was separately collected, and the activity of MPO was determined using a myeloperoxidase assay kit. The mRNA transcription levels of tight-junction proteins (ZO-1, Occludin, Claudin-2) in colon samples (0.1 g) from each mouse group were measured. Data were analyzed via the 2-△△Ct method. For protein analysis, colon tissues (1 g) from each group were used for protein extraction, followed by lysis and quantification. Following the normalization of protein quantities across all experimental groups, a Western blot assay was conducted. Mouse anti-ZO-1, anti-Occludin, and anti-Claudin-2 monoclonal antibodies served as primary antibodies, with horseradish peroxidase-labeled goat anti-mouse IgG as the secondary antibody (d). Bars represent the mean ± standard error value of each group (*P < 0.05, **0.01 < P < 0.05, ***P < 0.01 vs DSS, #P < 0.05, ##0.01 < P < 0.05, ###P < 0.01 vs pTE/27-2, n = 3, per group).
Figure 8. Effects of different treatments on colon length in colitis mice. Measure the length of the colon of each group of mice (a). Bars represent the mean ± standard error value of each group (b), (*P < 0.05, **0.01 < P < 0.05, ***P < 0.01 vs DSS, #P < 0.05, ##0.01 < P < 0.05, ###P < 0.01 vs pTE/27-2, n=3, per group).
Figure 9. Determination of inflammatory secretion level. Collect the serum samples from each group of mice on the 14th day. Measure the concentrations of pro-inflammatory factors (IL-6, IL-1β, TNF-α) and anti-inflammatory factor (IL-10) using the corresponding ELISA kits. Bars represent the mean ± standard error value of each group (*P < 0.05, **0.01 < P < 0.05, ***P < 0.01 vs DSS, #P < 0.05, ##0.01 < P < 0.05, ###P < 0.01 vs pTE/27-2, ns represents P > 0.05, n=3, per group).
- Section 3.7–IL-10 is anti-inflammatory cytokine while TNFa is pro-inflammatory. Revise, please.
Response: Thank you for your comments.
We apologize for our mistakes. We have meticulously revised the corresponding sections (Section 3.7, page15, lines 503-505) highlighted in yellow.
- Discussion should be improved:
- Advantages of paracasei pPG-SP-pTE/27-2 over L. paracasei (pPG/27-2) /L. paracasei pPG-SP-pTFF3/27-2 / L. paracasei pPG-SP-pEGF/27-2 should be discussed.
Response: Thank you for your comments. In the second paragraph of the Discussion (page 16, lines 549-553), the advantages of pPG-SP-pTE/27-2 (pPG-pTE/27-2) were elaborated upon. This particular section has been highlighted in yellow for easy identification.
“The fusion expression of pTE produced a greater area of scratch repair and a more rapid healing rate compared to pTFF3 or pEGF individually. The data indicated that pTE may have a synergistic impact on enhancing cell migration and tissue healing, as shown by this experiment.”
- - Limitation of the study should be clearly addressed (for example only BALB/c, purity of the stimulators, evidence on individual pTFF3 and pEGF in vivo effects are not provided…)
Response: We are extremely grateful for your valuable suggestion. We wholeheartedly appreciate your feedback and have incorporated the relevant section into the manuscript as presented below. (Section 6)
“6. Research limitations”
Selecting an appropriate model according to research requirements is of utmost importance, given that distinct models can yield diverse phenotypes. In this experiment, only BALB/c mice were employed, which presents certain limitations. Additionally, the protein concentration range employed in this study was relatively narrow, potentially overlooking effects at lower or higher concentrations. The protein used in this study had a limited purity level, which might have introduced contaminants that interfered with the accurate measurement of its biological activity, leading to potential over-or under-estimation of the protein's true function. This study marks the initial stage in exploring the characteristics of the target proteins. This study marks the initial stage in exploring the characteristics of the target proteins. Nevertheless, we recognize that in-vivo expression data is of crucial importance, and future research will be centered around designing and conducting relevant experiments for further exploration.
21.- Corresponding references have to be cited in “Klaas Vandenbroucke and colleagues successfully expressed TFF3 in Lactobacillus… pre-serve the integrity of the intestinal barrier, and enhance intestinal health.”
Response: Thank you very much for pointing out the important issue of the need to cite relevant literature. The reference that you suggested has been accurately added to the revised manuscript. The citation is located at Paragraph 2 of the Discussion (page 16, lines 535-538), with the annotation number [1]. Through this supplementary citation, we believe that the academic support of the paper has been further strengthened, and it can more effectively demonstrate our research viewpoints. Thank you again for your professional guidance. We look forward to receiving more valuable comments from you on the revised version.
- Caluwaerts S, Vandenbroucke K, Steidler L. AG013, a mouth rinse formulation of Lactococcus lactis secreting human Trefoil Factor 1, provides a safe and efficacious therapeutic tool for treating oral mucositis [J]. Oral Oncology, 2010, 46 (7): 564-570. DOI: 1016/j.oraloncology.2010.04.008.
22.- “Significantly, in comparison to pTFF3 and pEGF, the fusion protein pTE has a more substantial proliferative impact at reduced doses.” Taking into account that proliferation was evaluated by absorbance (Figure 5), this comparison could be made only if all cultures (all stimulators in all doses) were analyzed simultaneously. Please indicate in Section 2.5 if that was a case. Otherwise, sentence should be deleted.
Response:Thank you very much for your valuable comments.
We sincerely appreciate your careful review and the valuable suggestion to delete the sentence in question. We have thoroughly considered your advice and have removed the specified sentence from the manuscript.
- - It has to be clearly indicated which outcomes are common to Lactobacilus and which are species / strain – specific. For example, “… Moreover, recombinant Lactobacillus exerts a regulatory effect…” should be revised as “Moreover, Lactobacillus fermentum (MTCC-5898) exerts a regulatory effect…” or “… Moreover, recombinant Lactobacillus may exert a regulatory effect…”
Response: Thank you very much for your helpful comments.
- We have made modifications in the new manuscript and highlighted them in yellow.
- We have replaced it with Moreover, recombinant Lactobacillus may exert a regulatory effect on page 17, lines 588-589, highlighted in yellow.
- List of references has to be check for consistency of citation and accuracy (name of author in low case, spaces, there are some missing of authors / volume / start page No., etc.). Furthermore, I could not find reference 23 (and provided title does not imply that any synergism was studied).
Response: Thank you very much for your useful comments.
We sincerely apologize for the formatting errors in the references. We have meticulously carried out all the necessary revisions to rectify these issues.
The 23rd reference in the original manuscript has been replaced with the following reference (reference 7 in the new manuscript).
[7] Huynh E, Li J. Generation of Lactococcus lactis capable of coexpressing epidermal growth factor and trefoil factor to enhance in vitro wound healing [J]. Applied Microbiology & Biotechnology, 2015, 99 (11): 4667-77. DOI: 10.1007/s00253-015-6542-0.
Minor concerns:
- Affiliations – smaller font for a (College of Veterinary Medicine…)
Response: Thank you very much for your helpful comments.
We have adjusted the font size of the corresponding section in the new manuscript.
- Introduction, 1st paragraph - Oxazolidinone has to be replaced by oxazolone
Response: Thank you very much for your suggestion.
We have replaced oxazolidinone with oxazolone in the manuscript (page2, line 53).
- Introduction, 2nd paragraph – abbreviations EGF and TFF have to be introduced at this point and further constantly used through the entire Manuscript. This has to be applied for all abbreviations in the manuscript –check, please!!!
Response: Thank you very much for your helpful suggestion.
We have made the corresponding corrections, which are highlighted in yellow, in the 2ed paragraph of the Introduction (line 59). Additionally, we have consistently used its abbreviation throughout the subsequent content of the manuscript.
- In vitro / vivo have to be written in Italic
Response: Thank you very much for your suggestion. We have made corrections and highlighted them in yellow on lines 85 and 563 of the manuscript.
- Introduction, 4th paragraph – trihotropic factor? Please check
Response: Thank you very much for your suggestion.
We sincerely apologize for our mistake and have already included it in the Introduction, 4th paragraph corrected and highlighted in yellow (page2, line 86).
- Numerical value and unit have to be separated by space (for example 37°C has to be 37 °C).
Response: Thank you very much for your suggestion.
We are sincerely sorry for our mistakes and have already made corrections and highlighted them in yellow.
- Section 2.3 – working dilution has to be associated with specific antibody (i.e. manufacturer and catalog no. have to be provided for each antibody) as that relationship is not universal. In addition, composition of the diluent solution has to be provided (diluent? % skim milk in…?)
Response: Thank you very much for your valuable comments.
We have provided additional explanations in section 2.3, on lines 134 and 136.
- Section 2.3, 2nd paragraph – “resistance” and “Cmr” have to be deleted
Response: Thank you very much for your comments.
We have deleted “resistance” and “Cmr” in section 2.3, 2nd paragraph.
- Section 2.4 – … for assessing “expression stability” instead of “genetic stability”
Response: Thank you very much for your comments.
We agree with your suggestion and have replaced “expression stability” with the content in lines 164, 165 and 176 related to genetic stability, and these replacement contents have been highlighted in yellow for easy identification.
- Material & Methods – it has to be written in past tense instead of imperative (sections 2.4, 2.8, 2.9, 2.10)
Response: Thank you very much for your comments.
We sincerely apologize for the incorrect use of tenses in our manuscript. We have rectified these errors in the relevant sections, which have been highlighted in yellow for your easy reference. Thank you very much for your attention to detail and valuable feedback.
- Section 2.5, 2nd line – a dot (.) has to be deleted
Response: Thank you very much for your comments.
We have rectified these errors in the relevant sections. Thank you very much for your attention to detail and valuable feedback.
- Section 2.9 – “…Western and IP lysis buffer containing 1% PMSF” has to be reformulated. Buffer for Western blot containing 1 % proteases inhibitors (PMSF)? Composition of the solution has to be provided.
Response: Thank you very much for your comments.
We sincerely appreciate your perceptive comment on the unclear experimental procedure description in our manuscript. We recognize that the original presentation could have confused readers, and we're truly grateful for your feedback. To fix this, we've reorganized the relevant text for better logical flow (page 7, lines 279-281).
Once again, thank you very much for your comments and suggestions.

Reviewer 2 Report
Comments and Suggestions for Authors
REVIEW
Dear authors,
The use of genetically modified organisms for use in the health area has always been controversial, however, in recent times openness has been given to their possible use due to the scientific evidence of their benefits, probiotics are one of those microorganisms that due to their safety characteristics and multiple beneficial properties attributed can be used for improvement and in this way, combat infectious and non-infectious intestinal pathologies. The work addresses a potential therapeutic with the use of Lactobacillus paracasei modified to express the proteins Trefoil Factor 3 and Epidermal Growth Factor, which have been shown to contribute to the protection and repair of the gastrointestinal mucosa, which is affected in multiple pathologies such as Inflammatory Bowel Disease, which means an improvement in the evolution of the disease in patients.
Please amend the requested comments and submit the revision file.
- I consider that more information needs to be added about the Lactobacillus paracasei strain used in the work in section 1. Introduction, they mention its characteristics in general but do not refer to the strain used specifically, indicate any reference where it was previously used and the benefits found.
- In section 5 Analysis of the Activity of pTFF3 and In Vitro the number of IPEC-J2 cells used is missing, as well as the concentration of the proteins, please complement this section.
- Table 1: How did they choose the dose concentration to be administered in the treatment in the in vivo model? And what is the reason for administering it for 14 consecutive days?
- What is the amount of protein released in the intestine of 2x109 CFU mice?
- Is there anything in the intestine of mice that could affect the production of TFF3 and EGF proteins?
- Is the colonization time in the intestine of mice known by Lactobacillus paracasei used in the present work?
- Can the presence of Lactobacillus paracasei modified with proteins modify the abundance and diversity of the intestinal microbiota? It has been shown that the presence of some probiotic strains can stimulate some autochthonous microorganisms that also promote the production of metabolites, bacteriocins or proteins that stimulate and regulate the gastrointestinal barrier.
- Is there evidence that prolonged stimulation with TFF3 and EGF proteins can cause any damage to the host?
- Section 11 Statistical analysis is missing.
- Information needs to be added to the caption of Figure 4d.
- Figure 7b: arrows need to be added to indicate the damage or main structural changes in the histopathology of the intestine.
- Figure 8 shows 2 images, which are not described in the figure caption.
- Write the terms in cursive “in vivo”, “in vitro” and “ad libitum”.
- Write the name in cursive “Lactobacillus”.
- Section Conclusions is missing.
Please amend the requested comments and submit the revision file.
Author Response
Dear reviewer,
Thank you very much for having our manuscript entitled “Lactobacillus paracasei Expressing Porcine Trefoil Factor 3 and Epidermal Growth Factor: A Novel Approach for Superior Mucosal Repair” (vetsci-3519735) reviewed in a timely and professional manner and for giving us an opportunity to revise the manuscript. We have revised the manuscript carefully, and used Microsoft Word's built-in track changes function to highlight any changes we made (The content may experience changes when opened in WPS.). We tried best to properly answer the questions and suggestions and made revision in the paper according to the comments (changes in yellow). We hope this new version of article will meet the requirement for publication on Veterinary Sciences.
Sincerely,
Fangjie Yin
Replies to Reviewer #2:
- I consider that more information needs to be added about the Lactobacillus paracasei strain used in the work in section 1. Introduction, they mention its characteristics in general but do not refer to the strain used specifically, indicate any reference where it was previously used and the benefits found.
Response:
Thank you very much for your helpful comments and those comments are all valuable and very helpful for revising and improving our paper. We are sorry about that we did not describe Lactobacillus paracasei 27-2 clearly. We have inserted supplementary explanations in the preface of the manuscript, which have been marked in yellow (page 3, lines 103-106). Additionally, we have cited the following references [1].
Reference:
- Li F, Mei Z, Ju N. Evaluation of the immunogenicity of auxotrophic Lactobacillus with CRISPR-Cas9D10A system-mediated chromosomal editing to express porcine rotavirus capsid protein VP4 [J]. Virulence, 2022, 13: 1315-1330. DOI: 10.1080/21505594.2022.2107646.
- In section 5 Analysis of the Activity of pTFF3 and In Vitro the number of IPEC-J2 cells used is missing, as well as the concentration of the proteins, please complement this section.
Response: Thank you very much for your valuable comments.
We truly appreciate your valuable opinion. In response, we have added supplementary explanations in section 2.5, which have been prominently highlighted in yellow for easy identification (page 5, lines 192-195). The details of the supplementary information are presented as follows:
“The cell density was adjusted to 2.0 × 10⁵/mL during cell passage. Then, 100 μL of cell suspension was added to each well of a 96-well plate. Subsequently, the plate with the added cell suspension was incubated in a moist environment containing 5% CO₂ for 12 h. Following the removal of the culture medium, the wells were washed with PBS. Then, 100 µL of protein samples at different concentrations (0, 12.5, 25, 50, 100 ng/mL) were added to each well, with six replicates for each sample.”
- Table 1: How did they choose the dose concentration to be administered in the treatment in the in vivo model? And what is the reason for administering it for 14 consecutive days?
Response: Thank you very much for the helpful comments.
Thank you very much for your attention and feedback on the interpretation of the research results in our paper. We highly value the question you raised regarding the administration of dosage in the in vivo model.
- Firstly, in the group receiving oral administration of recombinant paracasei 27-2 strains, our previous research findings indicated that daily administration of 200 μL of paracasei 27-2 at a concentration of 2×10⁹ CFU to mice for 7 consecutive days was the optimal method for enhancing mucosal immunity [1-3].
Regarding the DSS group, numerous studies have indicated that 2% DSS modeling exhibits a high success rate and stability in mice. When mice consume a solution containing 2% DSS, the majority of them can successfully develop colitis within a specific time frame (typically 5-7 days), with relatively minor individual variations [4-6]. This characteristic renders the experimental results highly reproducible, enabling researchers to carry out subsequent experimental observations and data analysis more effectively, thereby enhancing the reliability of the research. An overly high concentration of DSS may trigger severe toxic reactions in mice, such as severe diarrhea, rapid weight loss, and even death, which can disrupt the experimental process. Conversely, if the concentration is too low, it becomes challenging to induce typical colitis symptoms.
- Previously, researchers in our laboratory conducted in vivo colonization experiments of paracasei 27-2 in mice and piglets, and the results showed that it could colonize well in the intestine after 7 days of oral administration. DSS induced acute colitis in mice lasted for 7 days [1].
References:
- Ma Rumeng, Zhao Yuliang, Ma Mingshuang, et al. Comparative study on the immune responses induced by the protective antigen S1 of porcine epidemic diarrhea virus expressed by different porcine-derived recipient bacteria [J]. Acta Veterinaria et Zootechnica Sinica, 2024, 55(05): 2090-2099. (in Chinese)
- Li F, Mei Z, Ju N. Evaluation of the immunogenicity of auxotrophic Lactobacillus with CRISPR-Cas9D10A system-mediated chromosomal editing to express porcine rotavirus capsid protein VP4 [J]. Virulence, 2022, 13: 1315-1330. DOI: 10.1080/21505594.2022.2107646.
- Li F, Zhao H, Sui L, et al. Assessing immunogenicity of CRISPR-NCas9 engineered strain against porcine epidemic diarrhea virus. Appl Microbiol Biotechnol. 2024;108 (1):248. DOI: 10.1007/s00253-023-12989-0
- Kim HJ, Jeon HJ, Kim JY. Lactiplantibacillus plantarum HY7718 Improves Intestinal Integrity in a DSS-Induced Ulcerative Colitis Mouse Model by Suppressing Inflammation through Modulation of the Gut Microbiota. Int J Mol Sci. 2024;25 (1): DOI: 10.3390/ijms25010575
- Nunes NS, Chandran P, Sundby M, et al. Therapeutic ultrasound attenuates DSS-induced colitis through the cholinergic anti-inflammatory pathway. Ebio 2019; 45: 495-510. DOI: 10.1016/j.ebiom.2019.06.033
- Simeoli R, Mattace Raso G, Lama A, et al. Preventive and therapeutic effects of Lactobacillus paracasei B21060-based synbiotic treatment on gut inflammation and barrier integrity in colitic mice. J Nutr. 2015;145 (6):1202-10. DOI: 10.3945/jn.114.205989
- What is the amount of protein released in the intestine of 2x109 CFU mice?
Response: Thank you very much for your comments.
In this study, the concentrations of EGF and TFF3 proteins in the supernatant and bacterial sediment of three recombinant strains were measured at 6 h, 10 h, 14 h, 18 h, 22 h, and 24 h, respectively. The results were presented in Table 4. Under ideal conditions, when mice were orally administered with recombinant paracasei 27-2 strains at this specific dose, approximately 0.4 µg of exogenous protein was expressed (lines 241-242).
- Is there anything in the intestine of mice that could affect the production of TFF3 and EGF proteins?
Response: Thank you very much for your comments.
In this study, the TFF3 and EGF employed are both endogenous bioactive proteins in the porcine intestine. The intestinal milieu harbors a plethora of substances that modulate the expression of EGF and TFF proteins. These substances engage in intricate crosstalk, synergistically regulating intestinal homeostasis and the reparative processes. A more profound elucidation of the underlying mechanisms of action of these substances holds the potential to facilitate the development of more efficacious therapeutic strategies for intestinal pathologies [1-4].
- Joosten, S. P. J., Zeilstra, J. MET Signaling Mediates Intestinal Crypt-Villus Development, Regeneration, and Adenoma Formation and Is Promoted by Stem Cell CD44 Isoforms. Gastroenterology, 2017, 153(4), 1040-1053. e4. DOI: org/10.1053/j. gastro.2017.07.008
- Black, D. D., Ellinas, H. Apolipoprotein Synthesis in Newborn Piglet Intestinal Explants. Pediatric Research, 1992, 32(5), 553–558. DOI: org/10.1203/00006450-199211000-00014
- Podolsky, D. K., Gerken, G. Colitis-Associated Variant of TLR2 Causes Impaired Mucosal Repair Because of TFF3 Deficiency. 2009, 209–220. DOI:org/10. 1053/ gastro.2009.03.007
- Chwieralski CE, Schnurra I, Thim L. Epidermal growth factor and trefoil factor family 2 synergistically trigger chemotaxis on BEAS-2B cells via different signaling cascades. Am J Respir Cell Mol Biol. 2004;31 (5):528-37. DOI: 10.1165/rcmb.2003-0433OC
- Is the colonization time in the intestine of mice known by Lactobacillus paracasei used in the present work?
Response: Thank you very much for your comments.
Indeed, the Lactobacillus paracasei 27-2 utilized in this study was isolated from the porcine intestine and preserved by our laboratory. In prior research [1], its biological properties, in-vitro adhesion capabilities, and in-vivo colonization characteristics have been comprehensively investigated (Oral administration for seven days can establish a large amount of colonization in the intestine). Given its outstanding stress tolerance and remarkable in-vivo colonization potential, this particular strain was deliberately chosen as the oral vaccine delivery vehicle for the present experiment (lines 531-532).
- Ma Rumeng, Zhao Yuliang. Comparative study on the immune responses induced by the protective antigen S1 of porcine epidemic diarrhea virus expressed by different porcine-derived recipient bacteria [J]. Acta Veterinaria et Zootechnica Sinica, 2024, 55(05): 2090 - 2099. (in Chinese)
- Can the presence of Lactobacillus paracasei modified with proteins modify the abundance and diversity of the intestinal microbiota? It has been shown that the presence of some probiotic strains can stimulate some autochthonous microorganisms that also promote the production of metabolites, bacteriocins or proteins that stimulate and regulate the gastrointestinal barrier.
Response: Thank you very much for your valuable comments.
Related research has indicated that the dysbiosis of the gut microbiota can be rectified, and the body's immune function can be restored through the transplantation of probiotics [1]. The Lactobacillus paracasei 27-2 employed in this study was isolated from the porcine intestine and preserved by our laboratory. It represents a natural probiotic indigenous to the porcine intestine. Lactic acid bacteria exert their regulatory effect on the intestinal mucosal microbial barrier primarily by secreting organic acids and bacteriocins. This secretion leads to a reduction in intestinal pH, inhibits the growth and reproduction of pathogenic microorganisms, and modulates the balance of the intestinal microbiota. For instance, Derrien et al. [2] discovered that lactic acid bacteria can ferment carbohydrates and amino acids to generate secondary metabolites, which provide nutrients for other microorganisms. This process increases the abundance of specific symbiotic bacteria and thereby influences the balance of the gut microbiota. Hailin Zhang successfully expressed porcine β defense factor in Lactobacillus paracasei 27-2, and after oral immunization of mice and piglets, the immune ability of the animals was improved (data not published).
- SANG Y, BLECHA F. Porcine host defense peptides: Expanding repertoire and functions [J]. Developmental & Comparative Immunology, 2009, 33(3): 334-43.
- DERRIEN M, VAN HYLCKAMA VLIEG J E T. Fate, activity, and impact of ingested bacteria within the human gut microbiota [J]. Trends in Microbiology, 2015, 23(6): 354-66.
- Is there evidence that prolonged stimulation with TFF3 and EGF proteins can cause any damage to the host?
Response: Thank you very much for your valuable comments.
It has been recognized that EGF and TFF act synergistically during the intestinal repair process. Nevertheless, the potential impacts of long - term combined stimulation by them remain to be elucidated. It is also worth noting that, as reported in references [1,2], overexpression of EGF and TFF in vivo might potentially lead to certain damage to the intestinal microenvironment. However, the majority of authoritative literature exerts a positive influence on restoration and other aspects [3-9].
- Chen CL, Mehta VB, Zhang HY. Intestinal phenotype in mice overexpressing a heparin-binding EGF-like growth factor transgene in enterocytes. Growth Factors. 2010; 28 (2):82-97. DOI: 10.3109/08977190903407365
- Dossinger V, Kayademir T, Blin N. Down-regulation of TFF expression in gastrointestinal cell lines by cytokines and nuclear factors. Cell Physiol Biochem. 2002; 12 (4): 197-206. DOI: 10.1159/000066279
- Bradford EM, Ryu SH, Singh AP. Epithelial TNF Receptor Signaling Promotes Mucosal Repair in Inflammatory Bowel Disease. J Immunol. 2017; 199 (5): 1886-1897. DOI: 10.4049/jimmunol.1601066
- Krishnan K, Arnone B, Buchman A. Intestinal growth factors: potential use in the treatment of inflammatory bowel disease and their role in mucosal healing. Inflamm Bowel Dis. 2011; 17 (1): 410-22. DOI: 10.1002/ibd.21316
- Yang Y, Lin Z, Lin Q.Pathological and therapeutic roles of bioactive peptide trefoil factor 3 in diverse diseases: recent progress and perspective[J].Cell Death & Disease, 2022, 13(1).DOI:1038/s41419-022-04504-6.
- Xiao, P., Ling, H., Lan, G., Liu, J. Trefoil factors: Gastrointestinal-specific proteins associated with gastric cancer. Clinica Chimica Acta, 2015; 450, 127–DOI: org/10.1016/j.cca.2015.08.004
- Grases-Pintó B,Torres-Castro P,Abril-Gil M. TGF-β2, EGF and FGF21 influence the suckling rat intestinal maturation. J Nutr Biochem. 2025;135:DOI:10.1016/j.jnutbio.2024.109778
- Huynh E, Li J.Generation of Lactococcus lactis capable of coexpressing epidermal growth factor and trefoil factor to enhance in vitro wound healing [J]. Applied Microbiology & Biotechnology, 2015, 99 (11): 4667-77. DOI: 1007/s00253-015-6542-0.
- Chinery R, Playford R J.Combined intestinal trefoil factor and epidermal growth factor is prophylactic against indomethacin-induced gastric damage in the rat.[J].Clinical Science, 1995, 88(4):401-403.DOI:1042/cs0880401.
- Section 11 Statistical analysis is missing.
Response: Thank you very much for your helpful comments.
We are extremely grateful for your valuable suggestion. We wholeheartedly appreciate your feedback and have incorporated the relevant section into the manuscript as presented below (Section 2.11). Yellow highlighting has been applied to lines 296-304 on page eight of the manuscript.
2.11. Statistical Analysis
All the experiments in this study were repeated three times. The data were statistically analyzed using the Two-way ANOVA method with the Graphpad Prism 8.0 software. As shown in the significant difference analysis of P-values: * P < 0.05, 0.01 < **P < 0.05 and ***P < 0.01, while ns represents P > 0.05, and # represents the same significance range as *.
Values indicate mean ± standard deviation (n = 3); bolded font indicates maximum values; different letters indicate significant differences between groups (P < 0.05); the same letters indicate non-significant differences (P > 0.05).
- Information needs to be added to the caption of Figure 4d.
Response: Thank you very much for your valuable comments.
We have added information to Figure 4 as shown below. And the replacement has been completed in the beginner's draft.
Figure 4. The growth characteristics and stability of recombinant paracasei 27-2 strains were analyzed. The recombinant paracasei 27-2 strains underwent continuous subculturing for 20 generations. Plasmids were extracted at intervals of every five generations, and PCR identification (a, b, c) was executed using the primers pPG-F/R. Subsequently, the bacterial proteins were processed. Western blot (e, f, g) was then performed, with mouse anti-Flag monoclonal antibody serving as the primary antibody and HRP goat anti-mouse IgG as the secondary antibody. The recombinant paracasei 27-2 strains were inoculated at an inoculation ratio of 1%. The cell suspension was collected at two-hour intervals. The viable cell count was carried out by means of the plate counting method (d). (n = 3, per group)
- Figure 7b: arrows need to be added to indicate the damage or main structural changes in the histopathology of the intestine.
Response: Thank you very much for your valuable comments.
We sincerely appreciate your valuable opinion and have duly made the corresponding modifications. The revised image is presented as follows and has been substituted in the new draft.
Figure 7. During the period of DSS modeling, each mouse was weighed daily, and its mental state was recorded. The mice were scored according to the DAI scoring table (Table 2) (a). After the completion of the modeling, a section of the colon (approximately 1 cm in length) near the rectum was excised from the mice of each group for H&E staining (b). The black arrow denotes the infiltration of inflammatory cells, the yellow arrow indicates crypt loss, the green arrow signifies the loss of glandular structure, and the blue arrow represents the loss of goblet cells. Effects of different treatments on colonic MPO activity (c). Approximately 0.1 g of the colon from each mouse in each group was separately collected, and the activity of MPO was determined using a myeloperoxidase assay kit. The mRNA transcription levels of tight-junction proteins (ZO-1, Occludin, Claudin-2) in colon samples (0.1 g) from each mouse group were measured. Data were analyzed via the 2-△△Ct method. For protein analysis, colon tissues (1 g) from each group were used for protein extraction, followed by lysis and quantification. Following the normalization of protein quantities across all experimental groups, a Western blot assay was conducted. Mouse anti-ZO-1, anti-Occludin, and anti-Claudin-2 monoclonal antibodies served as primary antibodies, with horseradish peroxidase-labeled goat anti-mouse IgG as the secondary antibody (d). Bars represent the mean ± standard error value of each group (*P < 0.05, **0.01 < P < 0.05, ***P < 0.01 vs DSS, #P < 0.05, ##0.01 < P < 0.05, ###P < 0.01 vs pTE/27-2, n = 3, per group).
- Figure 8 shows 2 images, which are not described in the figure caption.
Response: Thank you very much for your helpful comments.
We fully agree with your suggestion, which will greatly help us improve the quality of our manuscript. We have added the following information to the manuscript (page 15, lines 487-489, lines 498-499).
Figure 8. Effects of different treatments on colon length in colitis mice. Measure the length of the colon of each group of mice (a). Bars represent the mean ± standard error value of each group (b), (*P < 0.05, **0.01 < P < 0.05, ***P < 0.01 vs DSS, #P < 0.05, ##0.01 < P < 0.05, ###P < 0.01 vs pTE/27-2, n=3, per group).
- Write the terms in cursive “in vivo”, “in vitro” and “ad libitum”.
Response: Thank you very much for your helpful comments.
We sincerely apologize for the formatting errors. We have meticulously carried out all the necessary revisions to rectify these issues and highlighted them in yellow (line 85, line 177, line 555, line 564, line 689, line 738).
- Write the name in cursive “Lactobacillus”.
Response: Thank you very much for your comments.
We sincerely apologize for the formatting errors. We have meticulously carried out all the necessary revisions to rectify these issues and highlighted them in yellow.
- Section Conclusions is missing.
Response: Thank you very much for your helpful comments.
Your comments have been invaluable in enhancing the quality of our manuscript.
We have added a conclusion section in the new manuscript and highlighted in yellow (section 5, page 17, lines 603-609), as shown below:
“5. Conclusions”
In summary, the oral administration of pPG-pTE/27-2 has been demonstrated to alleviate DSS-induced colitis in mice. This treatment not only suppresses the inflammatory response and mitigates intestinal damage but also enhances the mucosal barrier function of intestinal epithelial cells by upregulating the expression of tight junction proteins. This research offers valuable insights into the treatment of IBD, thereby laying a solid foundation for further exploration in this field.
Once again, thank you very much for your comments and suggestions.

Reviewer 3 Report
Comments and Suggestions for Authors
The manuscript entitled “Lactobacillus paracasei Expressing Porcine Trefoil Factor and Epidermal Growth Factor: A Novel Approach for Superior Mucosal Repair”, authored by Fangjie Yin el al., represents an interesting contribution to the knowledge of the mechanisms involved in the repair and protection of the intestinal epithelium. and analyzes the effects on it of molecules that act on the mechanisms of proliferation, inhibition of apoptosis and migration of the intestinal epithelium.
In particular, the authors study the effects of trefoil factor 3 (TFF3) and epidermal growth factor (EGF) using recombinant strains de Lactobacillus paracasei expressing porcine TFF3 (pTFF3), porcine EGF (pEGF), and one fusion protein (pTE), respectively. These proteins are capable of promoting IPEC-J2 cell proliferation and migration.
The authors also perform a series of in vivo experiments, and after inducing colitis with dextran sodium sulfate, they administer recombinant strains of Lactobacillus paracasei, which reduces colitis and ameliorates mucosal integrity by improving intracellular junctions and acting on the inflammatory process.
The authors conclude that the recombinant Lactobacillus paracasei engineered to express pTFF3 and pEGF may be an intervention of interest to treat some intestinal pathologies.
In general, the manuscritp is well written, it is easy to read, the battery of tests that are carried out is very extensive and supports the results of the study.
However, I have detected some errors in the manuscript that need to be clarified since not all readers are familiar with the terminology. I suggest that authors take special care that when acronyms are used for the first time, it is made clear what they mean. For example, when they talk about IPEC-J2 cells they should clarify that they are intestinal porcine enterocytes isolated from the jejunum of a neonatal unsuckled piglet. They talk about DDS solution but do not specify that it is dextran sodium sulfate.
A major problem, but solvable, occurs in figures 3 and 6(a): almost nothing of what is to be shown is appreciated. Authors should show these results at higher magnifications.
Finally, in Figure 8, the word “execute” must be replaced and replaced by sacrifice; Animals cannot be executed because there is no sentence on them.
Author Response
Dear reviewer,
Thank you very much for having our manuscript entitled “Lactobacillus paracasei Expressing Porcine Trefoil Factor 3 and Epidermal Growth Factor: A Novel Approach for Superior Mucosal Repair” (vetsci-3519735) reviewed in a timely and professional manner and for giving us an opportunity to revise the manuscript. We have revised the manuscript carefully, and used Microsoft Word's built-in track changes function to highlight any changes we made (The content may experience changes when opened in WPS.). We tried best to properly answer the questions and suggestions and made revision in the paper according to the comments (changes in yellow). We hope this new version of article will meet the requirement for publication on Veterinary Sciences.
Sincerely,
Fangjie Yin
Replies to Reviewer #3:
- However, I have detected some errors in the manuscript that need to be clarified since not all readers are familiar with the terminology. I suggest that authors take special care that when acronyms are used for the first time, it is made clear what they mean. For example, when they talk about IPEC-J2 cells they should clarify that they are intestinal porcine enterocytes isolated from the jejunum of a neonatal unsuckled piglet. They talk about DDS solution but do not specify that it is dextran sodium sulfate.
Response:
Thank you very much for your helpful comments and those comments are all valuable and very helpful for revising and improving our paper. We sincerely apologize for the incorrect use of full names and abbreviations. We have made the necessary corrections and highlighted them in yellow (line 33, line 34).
- A major problem, but solvable, occurs in figures 3 and 6(a): almost nothing of what is to be shown is appreciated. Authors should show these results at higher magnifications.
Response: Thank you very much for your valuable comments.
We have made modifications to Figure 3 and Figure 6 with higher resolution and replaced them in the new draft. Modify as follows:
Figure 3. Identification of proteins expressed in recombinant paracasei 27-2 strains by IFA. After culturing the recombinant paracasei 27-2 strains for 16 h, the bacterial cell pellets were separately collected, washed, and then fixed on the glass slides. Mouse anti-Flag monoclonal antibody was used as the primary antibody, and goat anti-mouse IgG labeled with FITC was used as the secondary antibody. Subsequently, the cell nuclei were stained with DAPI, and the results were observed under an inverted fluorescence microscope.
Figure 6. The analysis of the migration effect about the target protein on IPEC-J2. Using the optimal concentrations of each protein for cell proliferation (pEGF: 50 ng/mL, pTFF3: 100 ng/mL, pTE: 25 ng/mL, pEGF+pTFF3: pEGF-50 ng/mL, pTFF3-100 ng/mL), the above-mentioned proteins were added respectively after the cell scratch assay. The healing area of the scratch was observed under an inverted fluorescence microscope at 0 h, 12 h, and 24 h. Bars represent the mean ± standard error value of each group (*P < 0.05, **0.01 < P < 0.05, ***P <0.01 vs pPG, #P < 0.05, ##0.01 < P < 0.05, ###P < 0.01 vs pTE, ns represents P>0.05, n=3 per group).
- Finally, in Figure 8, the word “execute” must be replaced and replaced by sacrifice; Animals cannot be executed because there is no sentence on them.
Response: Thank you very much for your valuable comments.
We are extremely grateful for your valuable suggestion. The corresponding section of the manuscript has been revised as follows:
Once again, thank you very much for your comments and suggestions.

Round 2
Reviewer 1 Report
Comments and Suggestions for Authors
At first, I would like to apologize deeply to you and authors for typo error I made unintentionally in the first report. It resulted in an inappropriate revision of the manuscript. The error was made in comment No. 3: “… After first mentioning, Lactobacillus paracasei shoud be written as paracasei (followed by strain 27-2, pPG-SP-pTFF3/27-2, pPG-SP-pEGF/27-2 or pPG-SP-pTE/27-2) throughout the entire manuscript…” where I omitted “L.” before paracasei. Hence, after first mentioning, Lactobacillus paracasei should be written as L. paracasei (followed by strain 27-2, pPG-SP-pTFF3/27-2, pPG-SP-pEGF/27-2 or pPG-SP-pTE/27-2) i.e. “L.” should be added before paracasei at all positions (paracasei could not where standalone).
Although manuscript vetsci-3519735 is improved, some questions remain open. I refer to my previous report in the following comments:
- Correction related to comment No. 2 is not adequate. In this particular case, probiotic function could be assigned to L. paracasei but not to TFF3 and EGF. Hence, probiotic functions of TFF3 and EGF cannot be enhanced. There is only a possibility that expression of TFF3 and EGF enhanced probiotic function of L. paracasei.
- Revision related to comment No. 5 opens new issue. Authors said that fusion protein was quantified using two assays which results were further combined (lines 160 – 161). The way of results’ combining should be explained (Was there significant difference between results of two assays? Mean value from two assay is calculated?).
- Why fractional precipitation of proteins was performed (line 182: … 20%, 40%, and 50%.) and which fraction is used for measurement of pEGF and TFF3? In addition, text in lines 183-187 should be checked – proteins were dialyzed against PBS, precipitated from PBS (by centrifugation; speed should be provided) and reconstituted in PBS? That seems contradictory. What was used for EGF and TFF3 measurement – supernatant obtained by centrifugation or pellet that is further reconstituted in PBS?
- It is acceptable to use unpurified protein for stimulation. In response to question No. 15 authors state that no proteins other than pTE were detected, while information on the method used for control of purity of pTFF3, pEGF and pTE used in in vitro assays is not provided (response to question No. 8). Although pTE/pEFG/pTFF3 could dominate in culture supernatant, presence of other proteins (for example constitutively secreted by bacteria, raised due to decay of bacterial cells, etc.) could not be completely excluded. Hence, information on the content of active component (pEFG, pTFF3 or pTE) in protein mixture (expressed as µg of pEFG/pTFF3/pTE per mg of total proteins) should be provided as other bacterial proteins might be present and promote proliferation as well. In line, data on proliferation stimulated by corresponding amount of proteins isolated from paracasei pPG/27-2 should be also provided at Figure 5. What was used for pPG stimulation in scratch assay (Figure 6)? Further, it should be pointed out in Section 3.4 and Discussion (lines 549-553) that pTFF3, pEGF and pTE were used in different concentrations (concentrations that provided maximal proliferation) in scratch assay.
- The term “genetic stability” remains in Section 2.4 of the revised manuscript. This term is not appropriate in the actual context. It should be replaced by “expression stability” (minor concern No. 9)
- Comment related to Table 1 (question No. 10) is not completely addressed. DSS treatment (2%DSS (Free access to water, days 8 to 14)) should be added in the column “Oral dosage, per mouse per day” for groups pPG-pTE/27-2 and pPG/27-2.
The English still needs improvement.
Author Response
Dear reviewer,
Thank you very much for having our manuscript entitled “Lactobacillus paracasei Expressing Porcine Trefoil Factor 3 and Epidermal Growth Factor: A Novel Approach for Superior Mucosal Repair” (vetsci-3519735) reviewed in a timely and professional manner and for giving us an opportunity to revise the manuscript. We have revised the manuscript carefully, and used Microsoft Word's built-in track changes function to highlight any changes we made (The content may experience changes when opened in WPS.). We tried best to properly answer the questions and suggestions and made revision in the paper according to the comments (changes in green). We hope this new version of article will meet the requirement for publication on Veterinary Sciences.
We sincerely appreciate the reviewer's meticulous review and constructive feedback. Following your suggestion, we have revised all instances of "paracasei" to "L. paracasei" throughout the manuscript to adhere to standard microbial nomenclature guidelines and ensure terminological consistency. These changes have been carefully implemented in the revised version of the manuscript.
Sincerely,
Fangjie Yin
Replies to Reviewer #1:
- Correction related to comment No. 2 is not adequate. In this particular case, probiotic function could be assigned to L. paracasei but not to TFF3 and EGF. Hence, probiotic functions of TFF3 and EGF cannot be enhanced. There is only a possibility that expression of TFF3 and EGF enhanced probiotic function of L. paracasei.
Response: Thank you very much for your comments. Your suggestions are indeed helpful for us. We have made adjustments to the literature, and your suggestions will be extremely helpful in enhancing the quality of our manuscript. In response to your suggestion, we have made corresponding modifications in the manuscript, which are highlighted in green (page 3, lines 120-123).
“In this study, Lactobacillus paracasei 27-2, a strain isolated from the intestinal tract of piglets, was employed as the host organism. To enhance its probiotic efficacy, the strain was genetically modified to express TFF3 and EGF, two bioactive molecules known to promote intestinal health and epithelial repair.”
- Revision related to comment No. 5 opens new issue. Authors said that fusion protein was quantified using two assays which results were further combined (lines 160 – 161). The way of results’ combining should be explained (Was there significant difference between results of two assays? Mean value from two assay is calculated?).
Response: Thank you very much for your helpful comments and those comments are all valuable and very helpful for revising and improving our paper. We are sorry about that we did not describe the procedures clearly (page 5, lines 179-183).
“The concentrations of pTFF3 and pEGF in the fusion protein were quantitatively determined using specific ELISA kits for each target protein (pTFF3 ELISA kit and pEGF ELISA kit, respectively). The experimental values were calculated by summing the mean values obtained from the pTFF3 and pEGF measurements. All experiments were performed in triplicate to ensure statistical reliability.”
- Why fractional precipitation of proteins was performed (line 182: … 20%, 40%, and 50%.) and which fraction is used for measurement of pEGF and TFF3? In addition, text in lines 183-187 should be checked – proteins were dialyzed against PBS, precipitated from PBS (by centrifugation; speed should be provided) and reconstituted in PBS? That seems contradictory. What was used for EGF and TFF3 measurement – supernatant obtained by centrifugation or pellet that is further reconstituted in PBS?
Response: Thank you very much for your helpful comments.
(1) Why fractional precipitation of proteins was performed (line 182: … 20%, 40%, and 50%.) and which fraction is used for measurement of pEGF and TFF3?
Response: Graded precipitation significantly enhances separation efficiency by allowing selective protein fractionation. In contrast, a single-step precipitation often results in non-specific co-precipitation of multiple proteins, which compromises the purity of the target protein. By implementing a gradual salt concentration gradient, graded precipitation minimizes protein co-precipitation and enables more precise separation of target proteins based on their differential solubility characteristics [1, 2]. Moreover, the gradual increase of salt concentration provides a milder condition compared to direct exposure to high-concentration salt solutions. This progressive approach is particularly advantageous for preserving the structural integrity and biological activity of proteins, as it minimizes the risk of denaturation that may occur during abrupt changes in ionic strength [3]. In this study, an initial ammonium sulfate saturation of 50% was employed to facilitate the selective precipitation and separation of pEGF and pTFF3 proteins. The relevant information has been incorporated into the revised manuscript, with all modifications clearly highlighted in green for easy reference (page 5, lines 204-207).
“The saturation levels were progressively increased in a stepwise manner, first to 20%, then to 40%, and finally to 50% saturation. By implementing a gradual salt concentration gradient, graded precipitation minimizes protein co-precipitation and enables more precise separation of target proteins based on their differential solubility characteristics”
[1] Mirica KA,Lockett MR,Snyder PW, et al. Selective precipitation and purification of monovalent proteins using oligovalent ligands and ammonium sulfate. Bioconjug Chem. 2012;23 (2):293-9. DOI: 10.1021/bc200390q
[2] Sookkumnerd, T., Hsu, J. T., Ito, Y. PURIFICATION OF PEG-PROTEIN CONJUGATES BY CENTRIFUGAL PRECIPITATION CHROMATOGRAPHY. Journal of Liquid Chromatography & Related Technologies. 2000; 23(13), 1973–1979. https://doi.org/10.1081/jlc-100100466
[3] Brgles M, Prebeg P, Kurtović T, et al. Optimization of tetanus toxoid ammonium sulfate precipitation process using response surface methodology. Prep Biochem Biotechnol. 2016;46 (7):695-703. DOI:10.1080/10826068.2015.1135452
(2) In addition, text in lines 183-187 should be checked – proteins were dialyzed against PBS, precipitated from PBS (by centrifugation; speed should be provided) and reconstituted in PBS? That seems contradictory. What was used for EGF and TFF3 measurement – supernatant obtained by centrifugation or pellet that is further reconstituted in PBS?
Response: We sincerely apologize for the oversight in clearly describing the experimental procedure. The relevant methodological details have now been explicitly stated in the revised manuscript, with the corresponding modifications clearly highlighted in green on page 5 (lines 210-212) for the reviewer's reference.
“Following dialysis, the sample was centrifuged at 10,000×g for 5 min at 4 °C to remove insoluble precipitates, and the resulting supernatant was carefully collected for subsequent analysis.”
- It is acceptable to use unpurified protein for stimulation. In response to question No. 15 authors state that no proteins other than pTE were detected, while information on the method used for control of purity of pTFF3, pEGF and pTE used in in vitro assays is not provided (response to question No. 8). Although pTE/pEFG/pTFF3 could dominate in culture supernatant, presence of other proteins (for example constitutively secreted by bacteria, raised due to decay of bacterial cells, etc.) could not be completely excluded. Hence, information on the content of active component (pEFG, pTFF3 or pTE) in protein mixture (expressed as µg of pEFG/pTFF3/pTE per mg of total proteins) should be provided as other bacterial proteins might be present and promote proliferation as well. In line, data on proliferation stimulated by corresponding amount of proteins isolated from paracasei pPG/27-2 should be also provided at Figure 5. What was used for pPG stimulation in scratch assay (Figure 6)? Further, it should be pointed out in Section 3.4 and Discussion (lines 549-553) that pTFF3, pEGF and pTE were used in different concentrations (concentrations that provided maximal proliferation) in scratch assay.
Response: Thank you very much for your helpful comments.
- It is acceptable to use unpurified protein for stimulation. In response to question No. 15 authors state that no proteins other than pTE were detected, while information on the method used for control of purity of pTFF3, pEGF and pTE used in in vitro assays is not provided (response to question No. 8). Although pTE/pEFG/pTFF3 could dominate in culture supernatant, presence of other proteins (for example constitutively secreted by bacteria, raised due to decay of bacterial cells, etc.) could not be completely excluded. Hence, information on the content of active component (pEFG, pTFF3 or pTE) in protein mixture (expressed as µg of pEFG/pTFF3/pTE per mg of total proteins) should be provided as other bacterial proteins might be present and promote proliferation as well.
Response: We have carefully revised the concentration-related descriptions for pEFG, pTFF3, and pTE throughout the manuscript to ensure accuracy and consistency. We have also conducted unit conversions in Tables 3 and 4 to ensure consistency and facilitate data interpretation. (page 6, lines 222-223, lines 234-236, Table 3, Table 4, page 12, lines 412-418, lines 431-435, lines 444-447, lines 461-464, page 17, lines 590-594).
“Table 3. Identification of pTFF3, pEGF and pTE expressed in supernatant of the recombinant L. paracasei 27-2 strains”
Incubation time |
Expression in supernatants of recombinant L. paracasei 27-2 strains pTFF3, pEGF and pTE (μg/mg) |
||
pTFF3 |
pEGF |
pTE |
|
6h |
0.15±0.02 |
0.10±0.02 |
0.07±0.03 |
10h |
0.35±0.01 |
0.17±0.02 |
0.19±0.04 0.30±0.05 0.37±0.05b 0.31±0.03 0.25±0.05 |
14h |
0.45±0.04 |
0.28±0.01 |
|
18h |
0.48±0.02a |
0.50±0.01a |
|
22h |
0.30±0.02 |
0.35±0.02 |
|
24h |
0.20±0.03 |
0.23±0.01 |
Note: The culture supernatants of the recombinant L. paracasei 27-2 strains were separately collected at cultivation durations of 6 h, 10 h, 14, 18 h, and 22 h. The data were presented as the mean ± standard deviation (n = 3). For different groups, the presence of distinct letters indicated a significant difference (P < 0.05), while the same letter indicated no significant difference between groups (P > 0.05).
“Table 4. Identification of pTFF3, pEGF and pTE expressed in cell lysates of the recombinant L. paracasei 27-2 strains”
Incubation time |
Expression in lysates of recombinant L. paracasei 27-2 strains pTFF3, pEGF and pTE (μg/mg) |
||
pTFF3 |
pEGF |
pTE |
|
6h |
0.18±0.02 |
0.16±0.01 |
0.12±0.01 |
10h |
0.35±0.01 |
0.28±0.02 |
0.27±0.03 0.45±0.01 0.90±0.07a 0.51±0.06 0.39±0.04 |
14h |
0.45±0.04 |
0.74±0.06 |
|
18h |
0.95±0.01a |
0.91±0.07a |
|
22h |
0.80±0.01 |
0.41±0.03 |
|
24h |
0.57±0.01 |
0.36±0.01 |
Note: The cell lysates of the recombinant L. paracasei 27-2 strains were separately collected at cultivation durations of 6 h, 10 h, 14, 18 h, and 22 h. The data were presented as the mean ± standard deviation (n = 3). For different groups, the presence of distinct letters indicated a significant difference (P < 0.05), while the same letter indicated no significant difference between groups (P > 0.05).
(2) In line, data on proliferation stimulated by corresponding amount of proteins isolated from paracasei pPG/27-2 should be also provided at Figure 5.
Response: We have carefully revised Figure 5 to enhance its clarity and scientific accuracy.
“Figure 5. The processed pTFF3, pEGF, and pTE proteins at varying concentrations (0, 12.5 ng/mL (5.68×10-3 µg/mg), 25 ng/mL (1.14×10-2 µg/mg), 50 ng/mL (2.27×10-2 µg/mg), 100 ng/mL (4.54×10-2 µg/mg), 200 ng/mL (9.08×10-2 µg/mg) and 400 ng/mL (1.82×10-1 µg/mg)), with the unit type specified elsewhere in the experimental context were dispensed into a 96-well plate populated with IPEC-J2 cells that had been pre-cultured for 12 h (100 μL per well). The pPG group represents the total protein concentration and serves as a control for normalization in our protein quantification experiments. Absorbance values at an optical density of 450 nm (OD450) were measured at 30-minute intervals. This measurement process was continued until the obtained values entered the optimal reading range defined by the CCK-8 kit protocol. For each protein sample, six replicate wells were set up, and the entire experimental setup was replicated three times in parallel for each group to ensure statistical robustness. Bars represent the mean ± standard error value of each group (*P<0.05, **0.01<P<0.05, ***P<0.01 vs 0 ng/mL, n=3 per group).”
(3) What was used for pPG stimulation in scratch assay (Figure 6)?
Response: We have supplemented and improved Figure 6 and the corresponding parts in the text. We have supplemented and enhanced Figure 6 as well as the relevant sections in the text. In the cell scratch experiments, the pPG group utilized a total protein concentration of 100 ng/mL, which was shown to optimize the cell proliferation ability (page 14, lines 461-463).
“Figure 6. The analysis of the migration effect about the target protein on IPEC-J2. Using the optimal concentrations of each protein for cell proliferation (pPG: 100 ng/mL (total protein concentration), pEGF: 50 ng/mL (2.27×10-2 µg/mg), pTFF3: 100 ng/mL (4.54×10-2 µg/mg), pTE: 25 ng/mL (1.14×10-2 µg/mg), pEGF+pTFF3: pEGF-50 ng/mL (2.27×10-2 µg/mg), pTFF3-100 ng/mL (4.54×10-2 µg/mg)), the above-mentioned proteins were added respectively after the cell scratch assay. The healing area of the scratch was observed under an inverted fluorescence microscope at 0 h, 12 h, and 24 h. Bars represent the mean ± standard error value of each group (*P < 0.05, **0.01 < P < 0.05, ***P <0.01 vs pPG, #P < 0.05, ##0.01 < P < 0.05, ###P < 0.01 vs pTE, ns represents P>0.05, n=3 per group).”
(4) Further, it should be pointed out in Section 3.4 and Discussion (lines 549-553) that pTFF3, pEGF and pTE were used in different concentrations (concentrations that provided maximal proliferation) in scratch assay.
Response: We have supplemented and improved in section 3.4 and the discussion section (page 13, lines 444-447, page 17, lines 590-594).
“Using the optimal concentrations of each protein for cell proliferation (pPG: 100 ng/mL (total protein concentration), pEGF: 50 ng/mL (2.27×10-2 µg/mg), pTFF3: 100 ng/mL (4.54×10-2 µg/mg), pTE: 25 ng/mL (1.14×10-2 µg/mg), pEGF+pTFF3: pEGF-50 ng/mL (2.27×10-2 µg/mg), pTFF3-100 ng/mL (4.54×10-2 µg/mg)), the above-mentioned proteins were added respectively after the cell scratch assay.”
“In the scratch test, pTFF3, pEGF, and pTE were tested at their optimal concentrations to promote cell proliferation (pPG: 100 ng/mL (total protein concentration), pEGF: 50 ng/mL (2.27×10-2 µg/mg), pTFF3: 100 ng/mL (4.54×10-2 µg/mg), pTE: 25 ng/mL (1.14×10-2 µg/mg), pEGF+pTFF3: pEGF-50 ng/mL (2.27×10-2 µg/mg), pTFF3-100 ng/mL (4.54×10-2 µg/mg)).”
- The term “genetic stability” remains in Section 2.4 of the revised manuscript. This term is not appropriate in the actual context. It should be replaced by “expression stability” (minor concern No. 9)
Response: Thank you very much for your valuable comments.
We sincerely apologize for this oversight and have carefully revised the corresponding content in Section 2.4 of the manuscript (page 5, lines 186-187, line 198).
- Comment related to Table 1 (question No. 10) is not completely addressed. DSS treatment (2%DSS (Free access to water, days 8 to 14)) should be added in the column “Oral dosage, per mouse per day” for groups pPG-pTE/27-2 and pPG/27-2.
Response: Thank you very much for your valuable comments.
We sincerely regret this oversight and have addressed it by incorporating the necessary additional information into the relevant section of Table 1. These modifications have been clearly highlighted in green to facilitate the reviewer's evaluation (page 7).
Group |
Oral dosage, per mouse per day |
Number |
PBS |
200 μL (Days 1 to 14) |
8 |
pPG-pTE/27-2 |
200 μL (2×109 CFU, days 1 to 14, 2%DSS (Free access to water, days 8 to 14)) |
8 |
pPG/27-2 |
200 μL (2×109 CFU, days 1 to 14, 2%DSS (Free access to water, days 8 to 14)) |
8 |
DSS |
2%DSS (Free access to water, days 8 to 14) |
8 |
“Table 1. Experiment design.”
Note: Meanwhile, from day 8 to day 14, each mouse in each group except for the PBS group was free to drink 2% DSS solution daily. Each mouse in the DSS group drank 200 μL of PBS per day for 1 to 7 days. Under ideal conditions, approximately 0.4 µg of exogenous protein was expressed at this specific dose.
Once again, thank you very much for your comments and suggestions.
Reviewer 2 Report
Comments and Suggestions for Authors
Dear authors,
The observations were addressed and duly justified, and are reflected in the article.
The abbreviation "L." should be used only in the phrase "recombinant L. paracasei 27-2," as it is incorrect to mention only the species.
Author Response
Dear reviewer,
Thank you very much for having our manuscript entitled “Lactobacillus paracasei Expressing Porcine Trefoil Factor 3 and Epidermal Growth Factor: A Novel Approach for Superior Mucosal Repair” (vetsci-3519735) reviewed in a timely and professional manner and for giving us an opportunity to revise the manuscript. We have revised the manuscript carefully, and used Microsoft Word's built-in track changes function to highlight any changes we made (The content may experience changes when opened in WPS.). We tried best to properly answer the questions and suggestions and made revision in the paper according to the comments (changes in green). We hope this new version of article will meet the requirement for publication on Veterinary Sciences.
Sincerely,
Fangjie Yin
Replies to Reviewer #2:
- The abbreviation "L." should be used only in the phrase "recombinant L. paracasei27-2," as it is incorrect to mention only the species.
Response: Thank you very much for your comments. We have made adjustments to the literature, and your suggestions will be extremely helpful in enhancing the quality of our manuscript. In response to your suggestion, we have made corresponding modifications in the manuscript, which are highlighted in green.
Once again, thank you very much for your comments and suggestions.